# Ferrielectricity controlled widely-tunable magnetoelectric coupling in van der Waals multiferroics

Qifeng Hu [1,9], Yuqiang Huang[1,9], Yang Wang [2,9], Sujuan Ding[3,9], Minjie Zhang[1,9], Chenqiang Hua[1], Linjun Li [4] ✉, Xiangfan Xu [5], Jinbo Yang [6], Shengjun Yuan [7], Kenji Watanabe [8], Takashi Taniguchi [8], Yunhao Lu [1] ✉, Chuanhong Jin [3] ✉, Dawei Wang [2] & Yi Zheng [1] ✉

The discovery of various primary *ferroic* phases in atomically-thin van der Waals crystals have created a new two-dimensional wonderland for exploring and manipulating exotic quantum phases. It may also bring technical break-throughs in device applications, as evident by prototypical functionalities of giant tunneling magnetoresistance, gate-tunable ferromagnetism and non-volatile ferroelectric memory etc. However, two-dimensional multiferroics with effective magnetoelectric coupling, which ultimately decides the future of multiferroic-based information technology, has not been realized yet. Here, we show that an unconventional magnetoelectric coupling mechanism inter-locked with heterogeneous ferrielectric transitions emerges at the two-dimensional limit in van der Waals multiferroic $CuCrP_2S_6$ with inherent anti-ferromagnetism and antiferroelectricity. Distinct from the homogeneous antiferroelectric bulk, thin-layer $CuCrP_2S_6$ under external electric field makes layer-dependent heterogeneous ferrielectric transitions, minimizing the depolarization effect introduced by the rearrangements of $Cu^+$ ions within the ferromagnetic van der Waals cages of $CrS_6$ and $P_2S_6$ octahedrons. The resulting ferrielectric phases are characterized by substantially reduced interlayer magnetic coupling energy of nearly 50% with a moderate electric field of $0.3\,V\,nm^{-1}$, producing widely-tunable magnetoelectric coupling which can be further engineered by asymmetrical electrode work functions.

The interplay and mutual control of multiple ferroic orders, in parti-cularly magnetism and ferroelectricity, in a single-phase system pro-vides a fascinating platform for both fundamental understanding on correlated phenomena and developing the next-generation non-vola-tile storage technologies[1]. Despite the tremendous success in classi-fying and revealing multiferroic mechanisms in paradigmatic bulk systems, viz. type-I multiferroics with nearly independent magnetism and ferroelectricity[2] and type-II multiferroics originated from Dzyaloshinskii-Moriya interaction and other types of magnetic order-ings such as collinear magnetic structures[3-5], the quest for a room-temperature multiferroic with efficient coupling between magnetism and ferroelectricity remains as the holy grail of the resurgent field. Recent studies on magnetism and ferroelectricity in two-dimensional (2D) van der Waals (vdW) crystals open unparalleled opportunities for multiferroic research[6-10], considering that abundant choices of ferroic atomic layers with single-crystal quality can be readily tuned by elec-trostatic field, strain, interfaces and spin-lattice interactions[11-14]. Equally important, the Lego combination of different 2D ferroic building blocks may lead to the finding of novel multiferroic phases beyond the prevailing spin-driven ferroelectric paradigm[1,15,16].

Beyond the framework of the isotropic Heisenberg model, various vdW bulk magnets with significant magnetocrystalline anisotropy have been re-explored at the 2D limit with fruitful results, e.g., 2D $CrI_3$, $CrCl_3$, CrSBr, and $CrPS_4$ with interlayer antiferromagnetic (AFM) coupling and intralayer ferromagnetic (FM) ordering, layer-dependent FM in $Cr_2Ge_2Te_6$, $CrBr_3$ and $Fe_3GeTe_2$, and emergent intralayer AFM systems of $MnPS_3$ and CrOCl[13,14,17–27]. Gate-tunable magnetic properties have also been demonstrated by electric controlled magnetic transistors of 2D $Cr_2Ge_2Te_6$ and memristive devices of $CrI_3$[11,28]. As an electric counterpart to magnetism, 2D ferroelectricity in atomically thin vdW crystals is also not rare, such as SnTe, $CuInP_2S_6$, $In_2Se_3$, $WTe_2$ and bilayer boron nitrides with divergent ferroelectric mechanisms[8,29–34]. Intriguingly, unlike bulk materials, 2D vdW systems tend to host multiple ferroic orders in one single phase, e.g., ferroelectricity and ferroelasticity predicted in monolayer group IV monochalcogenides[35,36], and ferromagnetism and ferroelasticity in monolayer $\alpha$-SnO[37]. Recently, the experimental implementation of a 2D magnetoelectric multiferroic was reported in $NiI_2$[9], however, the electrical control of magnetism, and the vice versa, remains elusive in this system.

Belonging to the large family of metal thio- and selenophosphate vdW crystals with rich ferroic phases[21,29,38,39], $CuCrP_2S_6$ (CCPS) is a unique choice of magnetoelectric multiferroics for exploring 2D magnetism-ferroelectricity interactions with experimentally confirmed coexistence of interlayer AFM and intralayer stripe antiferroelectric (AFE) in the bulk crystals[40–51]. As illustrated in Fig. 1a, CCPS crystals have a monoclinic lattice of the space group *Pc* (No. 7), in which a monolayer can be viewed as hexagonal vdW cages consisting of interconnected $CrS_6$ and $P_2S_6$ octahedrons. Below the Néel temperature ($T_N$) of ~31 K, the magnetic moments of $Cr^{3+}$ become ferromagnetically aligned within individual monolayers, which are further

AFM coupled in perpendicular to the vdW plane (Fig. 1b). On the other hand, the ferroelectric property of CCPS is rooted in $Cu^+$ ions, which are randomly distributed within the FM $CrS_6$-$P_2S_6$ cages. At 145 K, the crystal becomes AFE, when the adjacent Cu ions form striped AFE alignment driven by a double-well pseudo potential[41]. In the AFE state, each CCPS monolayer is characterized by a distinctive mirror symmetry along the *b*-axis, in perpendicular to the $Cu^+$ AFE stripe chains.

Here, we report an unconventional magnetoelectric coupling (MEC) interlocked with electric field tunable ferrielectric (FiE) transitions in 2D multiferroic CCPS, unveiling the rich promises of discovering highly tunable 2D MEC mechanisms fundamentally different from the bulk counterparts. Using magnetic tunneling junctions (MTJs), we demonstrate the electric field control of interlayer magnetic coupling strength, when $Cu^+$ ions rearrange within the $CrS_6$-$P_2S_6$ cages to form a layer-dependent heterogeneous FiE phase. The existence and continuous control of the FiE state are unambiguously proved by temperature- and polarization-dependent second harmonic generation (SHG) experiments, in which the latter unveils a fingerprinting six-fold to two-fold symmetry transition when thin-layer CCPS is driven from the AFE state into the FiE phase by an external bias voltage ($V_b$). The formation of the FiE phase enforces substantially lattice distortions for the $CrS_6$ octahedrons, which effectively modulate the interlayer superexchange coupling strength via the interlayer Cr-S-S-Cr pathways. This unique FiE-controlled MEC mechanism has a wide tunability, evident by a remarkable reduction in the interlayer magnetic coupling energy of nearly 50% with a moderate electric field of 0.3 V nm$^{-1}$.

## Results

Due to the insulating nature, we study the layer-dependent 2D magnetism, ferroelectricity and magnetoelectric properties of CCPS based

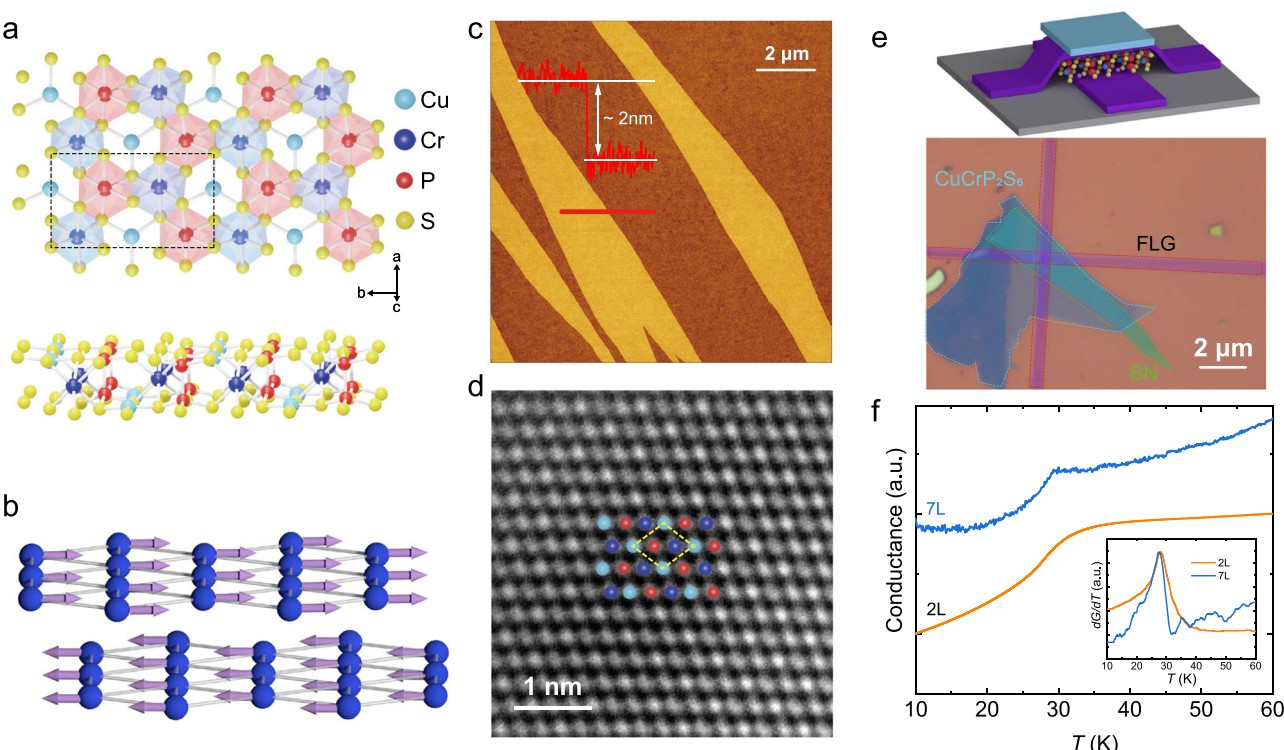

**Fig. 1 | Magnetoelectric multiferroics and Cu$^+$ ion polarizations in a ferromagnetic vdW cage. a** Top and side views of the lattice structure of a CCPS monolayer. Interconnected $CrS_6$ and $P_2S_6$ octahedrons form hexagonal vdW cages for polarizable $Cu^+$ ions. **b** Schematic of the interlayer AFM order in CCPS, where only the Cr atomic layers are shown for clarity. **c** NCAFM imaging and line profiling of bilayer CCPS flakes on $SiO_2$/Si substrates. **d** ADF-STEM image of a BL CCPS, showing high crystal quality without noticeable defects. **e** Top: device structure of thin-layer CCPS MTJs. Bottom: false-color optical image of a representative BL MTJ BL-S44. **f** Tunneling conductance *G* vs *T* curves of representative BL-S44 and 7L-S21. Inset: the corresponding d*G*/d*T*, revealing a thickness-independent AFM phase transition at about 30 K.

on vertical magnetic tunnel junctions, which are consisting of few-layer graphene (FLG) electrodes and thin layers of CCPS channels with consecutive thicknesses from bilayer (BL) to 10L (see Methods and Supplementary Information Note 1). Figure 1c shows the non-contact atomic force microscopy (NCAFM) of a BL CCPS sample, as evident by a 2 nm step height. Atomic-resolution annular dark-field scanning transmission electron microscopy (ADF-STEM) confirms the high crystal quality of exfoliated thin-layer CCPS samples with no apparent vacancies (Fig. 1d and Supplementary Note 1). If not specified otherwise, all tunneling results were measured at a base temperature of $T = 1.6$ K.

To investigate the magnetic phase transitions at the 2D limit, we have carried out $T$-dependent zero-bias tunneling experiments. As shown in Fig. 1e, f, using a small excitation current of 10 nA and zero magnetic field ($H$), the tunneling conductance of CCPS MTJs with different thicknesses consistently show an anomaly at ~ 30 K, in excellent agreement with the bulk $T_N$. The results strongly suggest that atomically thin CCPS retain the bulk AFM order, as expected for a vdW magnet with strong magnetic anisotropy. Using $H$-dependent tunneling spectroscopy, we observe the typical metamagnetic transition behavior for thin-layer CCPS MTJs, when interlayer Cr spin moments become symmetrically tilted along the $H$ direction to form the canted AFM (CAFM) state[14,24]. As shown in Fig. 2a, the tunneling current ($I_t$) of BL-S44 first increases smoothly as a function of in-plane $H_\parallel$, manifesting the spin-canting angle vs $H$ evolution of the CAFM phase. In the AFM ground state, two CCPS MLs with opposite magnetization direction form a spin filter tunneling barrier equivalent for electron transmission with opposite spins, as schemed in Fig. 2b (see Supplementary Note 2 for detailed information on the spin-filter mechanism). This produces a low $I_t$ which corresponds to a high resistance state[52–54]. When $\mu_0 H_\parallel$ reaches 3.2 T, the atomic spin valve is switched to the low

resistance state due to parallel magnetization between two adjacent MLs. The resulting $I_t$ becomes significantly enhanced attributed to asymmetric tunneling barriers for opposite spins[24,53]. For simplicity, we will refer this magnetization saturation state as the field-aligned ferromagnetic phase (FM′), and the saturation magnetic field as $H_{sat}$.

Such an $H$-driven CAFM to FM′ transition is further confirmed by $T$-dependent tunneling magnetoresistance (TMR), defined by TMR = $[I_t(H) − I_t(0)]/I_t(0) \times 100\%$[14,24,53]. As summarized in Fig. 2c, the CAFM-FM′ transitions gradually shift to lower values of $H_{sat}$ due to increased thermal fluctuation. Above $T_N$, the TMR curves exhibit a negative growth rate, in consistent with a paramagnetic state. It is illuminating to plot the field derivative of TMR ($dR/dH_\parallel$) into a 2D contour image as a function of both $T$ and $H_\parallel$, which clearly reveals the magnetic phase diagram of CCPS at the 2D limit (Fig. 2d). For CCPS MTJs with odd-layer thicknesses, for example a quintuple-layer MTJ (5L-S5), the corresponding $T$-dependent TMR and the 2D contour plot are shown in Fig. 2e, f, respectively. The overall phase diagram is agreeing with the BL device, indicating the same CAFM to FM′ phase transitions. The distinctive odd-layer effect below 2 T is due to a field-induced perfect collinear AFM state which is first reported in few-layer CrCl$_3$ MTJs[24]. Most critically, for an easy-plane magnetic anisotropy dominated vdW AFM, $H_{sat}$ is directly proportional to the interlayer exchange coupling energy $J_\perp$[24]. Compared to CrCl$_3$ with the same thickness, $H_{sat}$ of CCPS is nearly three times higher, which is crucial for the observation of the FiE-interlocked MEC mechanism.

Remarkably, CCPS MTJs show $V_b$ polarity-dependent $H_{sat}$, which reaches a prominent change of 50% with a moderate electric field (**E**) of 0.3 V nm$^{-1}$ for octuple-layer CCPS. As shown in Fig. 3a for MTJ 8L-S20, the normalized TMR curves exhibit an $H_{sat}$ value of 5.9 T for $V_b = − 1.2$ V, which drastically changes to 8.8 T for $V_b = 1.2$ V (see Supplementary Note 3 for TMR normalization procedures). Distinct from a

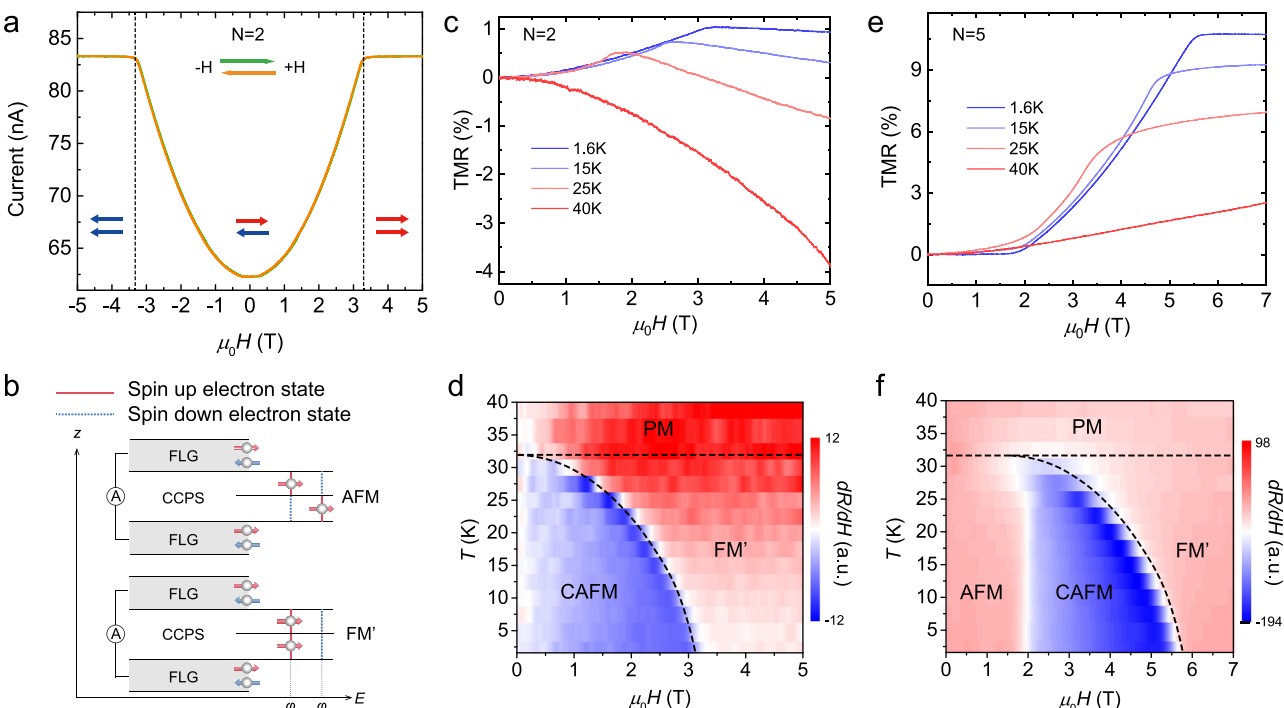

**Fig. 2 | Layer-dependent metamagnetism and even-odd effects in thin-layer CCPS. a** Tunneling current $I_t$ vs in-plane magnetic field $H_\parallel$ characteristics of a representative bilayer MTJ (BL-S44). BL-S44 exhibits a CAFM saturation field of 3.3 T, which is typical for BL devices. Here, green and orange arrows indicate the $H_\parallel$ ramping directions. **b** Illustration of the multiple spin-filter effect in vdW magnets with interlayer AFM ordering. **c** $T$-dependent TMR vs $H_\parallel$ of BL-S44, showing the dwindling of the CAFM saturation field when approaching $T_N$. **d** 2D contour plotting

of the first derivative of TMR ($dR/dH$) vs $H_\parallel$ and $T$ for BL-S44. By taking the derivative, the magnetic phase diagram of BL CCPS is clearly revealed. **e** $T$-dependent TMR vs $H_\parallel$ of quintuple-layer MTJ (5L-S12), showing the same dwindling of $H_{sat}$ when approaching $T_N$. **f** 2D contour of $dR/dH$ vs $H_\parallel$ and $T$ for 5L-S12. Below 2 T, the magnetic phase diagram of odd-layer CCPS is characterized by a low-field perfect collinear AFM zone, i.e., the odd-layer effect.

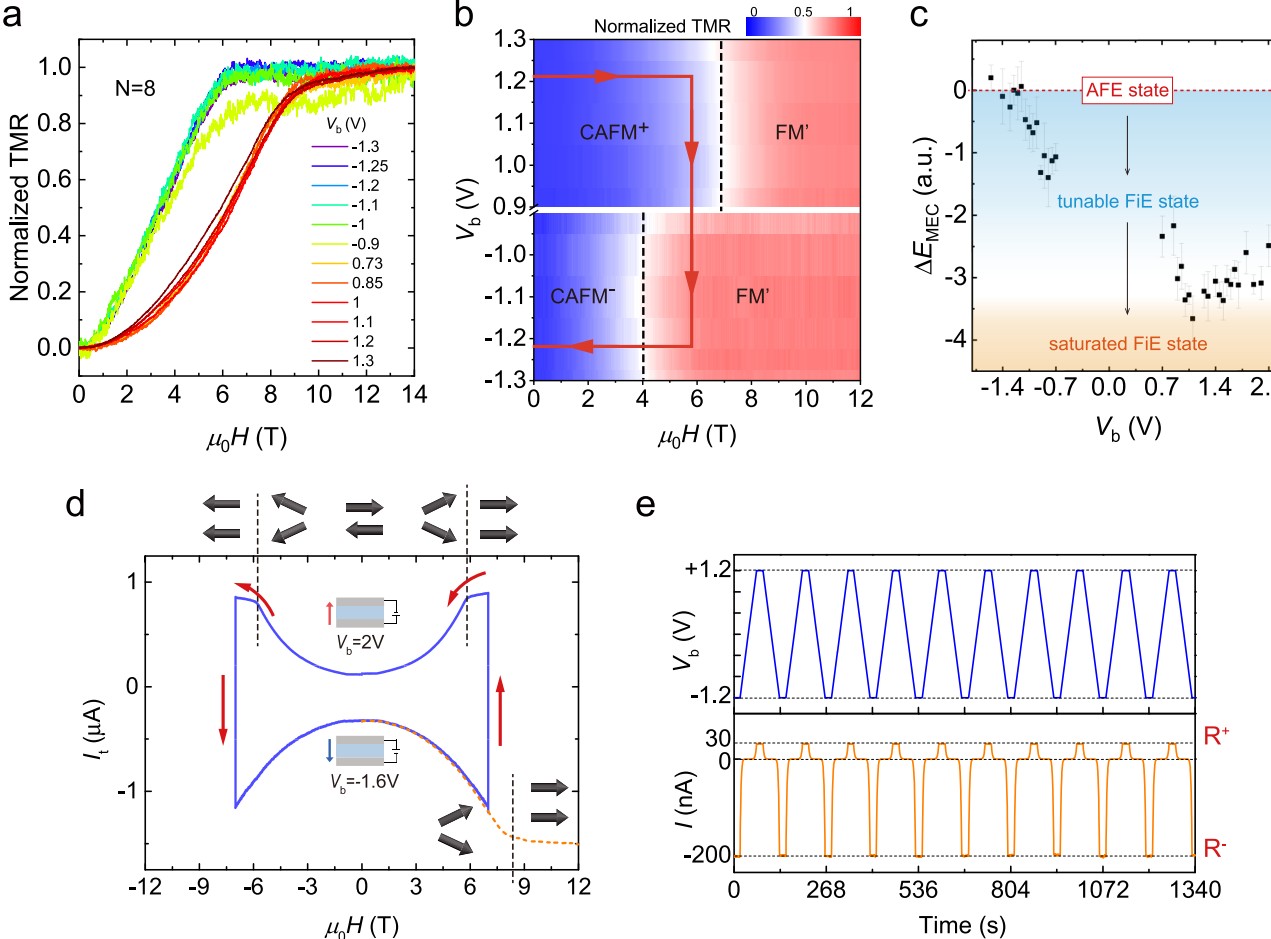

**Fig. 3 | Ultra-wide $V_b$-tunability of magnetoelectric coupling in thin-layer CCPS.**
**a** Normalized TMR vs $H_\parallel$ curves of MTJ 8L-S20 under different $V_b$ setpoints. **b** 2D contour plot of the normalized TMR vs $V_b$ and $H_\parallel$ in (**a**). It is noteworthy that for each $H_\parallel$ setpoint, the CAFM canting angle $\theta^\pm$ for opposite-polarity $V_b$ are divergent (see Supplementary Fig. 5 for CAFM $\theta^\pm$ calculations). **c** $\Delta E_{MEC}(\mathbf{E})$ vs $V_b$, extracted from the TMR data of 10L-S39, and the error bars are defined as the $\pm10\%$ variation

of half-maximum points in $dG/dH$ curve (see Supplementary Figs. 5 and 6).
**d** Dichotomic $I_t$ vs $H_\parallel$ characteristics of 8L-S20. Note that $I_t$ is symmetric on $H_\parallel$ ramping, excluding any ferroelectric polarization reversal. **e** Zero-$H$ $R_0^\pm$ reversals by switching the $V_b$ polarities. It is apparent that the $I_t - V_b$ sweeping is completely free of hysteresis.

paraelectric dielectric response, the $V_b$-dependence is conspicuously non-linear, which can be divided into two opposite-polarity regimes of 0.73 to 1.3 V for high $H_{sat}^+ = 8.8$ T and −0.9 to −1.3 V for low $H_{sat}^- = 5.9$ T, respectively. The dichotomic polarity behavior in $V_b$ and the wide tunability of $H_{sat}$ are best visualized by a 2D contour plot of TMR vs both $V_b$ and $H$. As shown in Fig. 3b, the large $H_{sat}$ window between positive and negative $V_b$ polarities opens a new possibility to directly manipulate the magnetic state of CCPS MTJs by $V_b$, i.e., the electrical control of magnetism. Indeed, by sweeping $V_b$ and $H$ to follow the path indicated by red solid lines in Fig. 3b, we can readily enforce a FM′ to CAFM phase transition, and the vice versa, simply by reversing the $V_b$ polarities. Since $H_{sat}$ represents the interlayer coupling energy $J_\perp$, we can directly measure the change of MEC energy introduced by $V_b$-polarity reversal by $\Delta E_{MEC} \propto (H_{sat}^+ - H_{sat}^-)$. Detailed TMR study reveals that in between $H_{sat}^\pm$, $H_{sat}$ is continuously tuned by $V_b$ setpoints before reaching the binary saturation values (Fig. 3c and detailed discussions in Supplementary Note 4). The large $\Delta E_{MEC}$ of ~1.08 meV per unit-cell (UC) explains the bistable magnetoresistance states TMR$^\pm$ for $H \neq 0$ (Fig. 3d). However, within the same AFM ground state, zero-$H$ resistance states $R_0^\pm$ remain binary valued, which can be reproducibly switched by reversing the $V_b$ polarities, as shown in Fig. 3e for MTJ 10L-S39.

Such an unexpected reversal behavior in $R_0^\pm$ reminds us the importance of the inherent stripe-AFE phase of CCPS, in analogy with

the resistance switching of graphene-ferroelectric field-effect transistors (GFeFETs) under a constant electrostatic doping[55,56]. However, unlike the hysteretic two-state switching in GFeFETs controlled by ferroelectric polarization reversals[55,56], the stripe-AFE phase of CCPS requires an enormous energy to be fully polarized, which can be evaluated quantitatively by the first-principle density-functional theory (DFT) calculations. As shown in Fig. 4a, an FE configuration for a BL UC will unrealistically increases the system energy by 337 meV per UC when compared with the stripe-AFE ground state (see Method for the calculation details). From the thermodynamic point of view, the FE state is also not favored due to the maximization of the depolarization effect, as depicted by the black hollow arrows. Based on DFT calculations, it is also not feasible to locally flip an AFE stripe by $\mathbf{E}$ within a single vdW ML. For BL CCPS, the lowest metastable state is to flip the anti-parallel stripes of the bottom ML toward the vdW interface, corresponding to a largely reduced energy requirement of 95 meV per UC (2L-mAFE-I). In Fig. 4b, we plot the energy potential diagram of BL CCPS as a function of the displacement ($\Delta d$) of the anti-parallel Cu-ion stripes within the bottom ML, suggesting a rapid growth in the system energy by approaching the Cr atomic plane. Note that even with the presence of an impractical $\mathbf{E}$ of 2 V nm⁻¹, the energy of 2L-mAFE-I remains substantially higher than the AFE ground state, and the energy barrier in-between is far too high to be crossed by Cu⁺ ions (Supplementary Note 5).

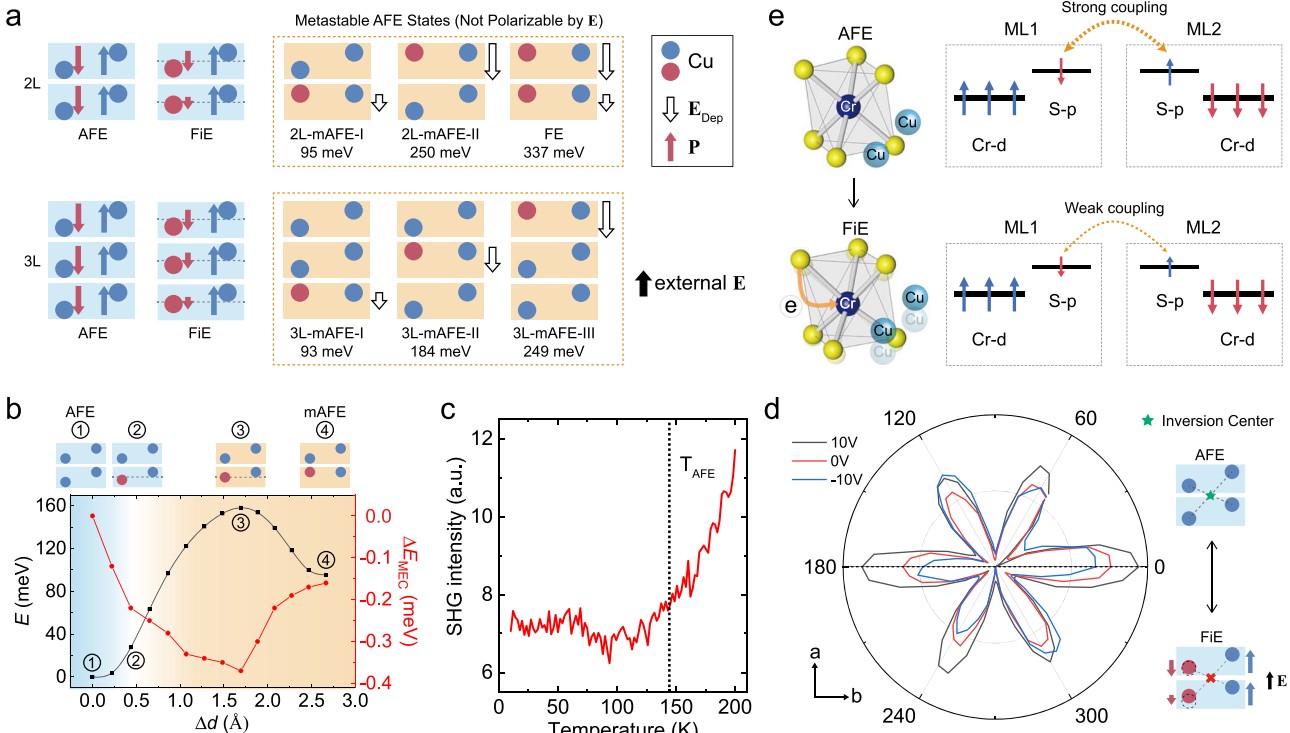

**Fig. 4 | Ferrielectricity controlled magnetoelectric coupling in thin-layer CCPS. a** Lattice model representation of the stripe-AFE state (first column), the **E**-enforced heterogenous FiE phase (second column), and energy unfavorable metastable AFE states (third column; grouped in orange dashed lines) for BL and trilayer CCPS, respectively. Here, the vdW monolayer cages are represented by blue/orange rectangles, standing for energy accessible and inaccessible respectively. Within the cages, the ground-state (polarized) Cu⁺ ions are depicted by blue (red) circles. Here, 2L-mAFE-I and 2L-mAFE-II correspond to the flipping of an anti-parallel Cu-ion stripe of within the bottom and top ML, respectively. **b** Black curve: Energy potential diagram vs $\Delta d$ (anti-parallel Cu⁺ displacement) of BL CCPS from the AFE ground state to 2L-mAFE-I. Red curve: $\Delta E_{MEC}$ on $\Delta d$, which monotonically decreases when Cu⁺ ions move towards the Cr atomic plane. **c** SHG intensity vs $T$, showing the AFE transition and the frozen of Cu⁺ ions at around 140 K. **d** Polarization-dependent SHG at 77 K under different $V_b$. The distinctive six-fold to two-fold transition in the polarization-SHG patterns proves that the non-linear optical anisotropy of thin-layer CCPS associated with inversion symmetry breaking can be continuously tuned by $V_b$. **e** Illustration of the FiE-interlocked MEC mechanism in CCPS, in which lattice distortions introduced by Cu⁺ ion displacements efficiently lower the interlayer superexchange interaction.

Indeed, for all CCPS MTJs, the $I_t − V_b$ characteristics measured with various $H$ setpoints (Supplementary Note 6) are thoroughly free of hysteresis behaviors, in consistent with the DFT calculations that predict no metastable phase transition under realistic **E** values. The DFT-based analyses of flipping a minimal AFE stripe can be readily extended to different ML numbers, as shown in the bottom panels of Fig. 4a for trilayer CCPS (quadruple-layer in Supplementary Note 5). With increased layer numbers, different metastable states can be used as an ML index of the flipped AFE stripe, whose energy cost is closely related to a peculiar layer-dependent depolarization mechanism. Using BL CCPS as an example, by flipping one AFE stripe in the top ML, the energy requirement of 2L-mAFE-II is drastically increased by nearly four times to 250 meV per UC, when compared with 2L-mAFE-I. The large energy difference between 2L-mAFE-I and 2L-mAFE-II is due to the fact that, for the former, the local flipped AFE domain is effectively screened by the neighboring AFE ML to minimize the depolarization field. Such a unique dipole screening mechanism for vdW ferroelectrics, designated as the vdW depolarization effect hereafter, can be well reproduced by DFT calculations of layer-dependent charge density distribution in response to AFE stripe displacements (see Supplementary Note 5 for details).

Although the metastable AFE states are not energy accessible by experimental **E**, the CCPS lattice spontaneously responds to the external electric field by readjusting the positions of Cu⁺ ions within the vdW cages of CrS₆ and P₂S₆ octahedrons. Our DFT calculations reveal that, the polarizations are predominated by layer-dependent rearrangements of the anti-parallel Cu⁺ ions, leading the formation of a heterogenous FiE state. As schemed in the second column of Fig. 4a, due to the unique vdW depolarization effect, anti-parallel Cu⁺ ions in different MLs will move toward the Cr atomic plane under **E**, with $\Delta d$ proportional to the energy of the ML-indexed metastable AFE states. The formation of an **E**-dependent FiE state is confirmed by SHG experiments, with tunable $V_b$ applied to the heterostructures of FLG/BN/CCPS/BN/FLG. As shown in Fig. 4c, by cooling down a 6L device, the $T$-dependent SHG intensity decreases monotonically and eventually stabilizes at about 140 K, which manifests the AFE transition of thin-layer CCPS when Cu⁺ ions lock into the stripe-AFE positions. We also conduct polarization-dependent SHG measurements at 300 K and 77 K, respectively, with different $V_b$ setpoints. Strikingly, polarization-SHG at 77 K unveils a fingerprinting six-fold to two-fold symmetry transition when thin-layer CCPS under **E** is driven from the AFE state into the FiE phase (Fig. 4d). In stark contrast, the same CCPS device at 300 K consistently shows a nearly six-fold symmetry for different $V_b$ (Supplementary Fig. 14c). Similar results are well reproduced on multiple devices, and also repeated on an FLG/CCPS/SiO₂/Si structure, in which the heavily doped silicon substrate was used as the bias electrode. The $V_b$-dependent SHG intensity reflects continuous inversion symmetry breaking induced by **E**. As illustrated in the insets of Fig. 4d, for the stripe-AFE ground state, there exists an inversion center between two neighboring monolayers, while the $V_b$-driven heterogeneous FiE state has reduced inversion symmetry due to layer-dependent rearrangements of anti-parallel Cu⁺ ions within the FM vdW cages.

Most unusually, the formation of **E**-dependent heterogenous FiE state is accompanied by a significant reduction in the magnetic energy difference between the interlayer FM′ and AFM configurations, which manifests the change of $V_b$-tunable MEC energy $\Delta E_{MEC}$. Using the illustrative example of BL CCPS, the DFT calculations deduce that the FiE state has monotonically decreased $\Delta E_{MEC}(\mathbf{E})$, defined by $\Delta E_{MEC}(\mathbf{E}) = (E_{FM'}(\mathbf{E}) - E_{AFM}(\mathbf{E})) - (E_{FM'}(0) - E_{AFM}(0))$, when **E** drives anti-parallel positioned $Cu^+$ ions toward the energy saddle point of the Cr atomic plane (see Supplementary Note 7 for the detailed formulation of $\Delta E_{MEC}(\mathbf{E})$). As illustrated in Fig. 4e, for thin-layer CCPS, the interlayer superexchange that determines $J_\perp$ is intermediated by the interfacial majority spin $p$-orbitals of S atoms, which are antiparallel to the Cr magnetic moments due to the formation of Cr-S valence bonds[57]. The stripe-AFE ground state corresponds to a relative symmetric $CrS_6$ octahedron with minimal lattice distortions (Fig. 4e and DFT calculations in Supplementary Table 1). In response to external electric field, anti-parallel $Cu^+$ ions move within the FM vdW cages, enforcing substantially lattice distortions for the $CrS_6$ octahedrons by stretching the average Cr-S bond lengths. The resulting attenuation of the Cr-S bonding energy effectively decreases the net majority spins of S atoms, which changes from 0.135 electrons per S for $d = 0$ Å (the equilibrium stripe-AFE position) to 0.092 electrons per S for $\Delta d = 0.44$ Å, and thus, efficiently lowers the interlayer AFM coupling.

As summarized in Fig. 4b, the collective movements of anti-parallel $Cu^+$ ions within the FM vdW cages reach the maximum $\Delta E_{MEC}$ of $-0.37$ meV for the saddle point position $\Delta d = 1.69$ Å. It should be noted that $\Delta E_{MEC}(\mathbf{E})$ has a maximized change rate within the **E**-polarizable regime, which is highlighted by the blue color ($\mathbf{E} < 1.5$ V nm$^{-1}$; see Supplementary Note 6 for electric breakdown test). In this polarization regime, $\Delta E_{MEC}$ rapidly reaches $-0.15$ meV for a minute $\Delta d$ of 0.29 Å, which is qualatatively in agreement with the experimental $\Delta E_{MEC}$ estimated by $H_{sat}^{\pm}$. As a direct consequence of this ultra $V_b$-tunable $\Delta E_{MEC}$, the $V_b$-dependent heterogenous FiE state has drastically reduced $J_\perp$, which changes by nearly 50% with a moderate **E** of 0.3 V nm$^{-1}$ as shown in Fig. 3c. After elucidating the $V_b$-tunable MEC mechanism, the final key element is to understand the $V_b$-polarity asymmetry in the MEC coupling, which is attributed to a build-in potential introduced by the asymmetric work functions of top and bottom FLG electrodes. Using Fowler-Nordheim (FN) tunneling in complementary with scanning Kelvin probe microscope (SKPM), we determined that the asymmetric work functions can readily reach 0.5 eV due to the Dirac electronic structure of graphene and the interfacial doping by AFE CCPS flakes[55,56] (see Supplementary Note 4 and Note 8 for detailed analyses using FN tunneling and SKPM). As shown in Supplementary Fig. 7a, the existence of a build-in potential makes the two otherwise $V_b$-polarity equivalent FiE states asymmetric in energy, in analogy to GFeFETs under a constant electrostatic doping[55,56].

## Discussion

The striking energy difference between the 2L-mAFE-I and 2L-mAFE-II states of BL CCPS provide an amazing example on how the well-established depolarization mechanism may become peculiarly different at the 2D limit in a vdW multiferroic. The discovered FiE interlocked widely-tunable MEC mechanism also suggests the great potential of realizing highly-efficient electric control of 2D magnetism in emergent vdW multiferroics. Equally important, the multiferroicity in thin-layer CCPS opens the possibility for the investigation of 2D quantum phase transitions, and the fabrication of complex vdW heterostructure for the realization of two-phase multiferroic for novel electronically controlled spintronics and Josephson junctions.

## Methods
### Crystal growth
Bulk CCPS single crystals were synthesized by the standard self-flux method. High purity Cu (99.999%), Cr (99.996%), P (99.999%) and S

(99.5%) powders were mixed stoichiometrically and loaded into a quartz tube in an argon-filled glove box. The quartz tube was evacuated to a pressure of $10^{-2}$ Pa before flame-sealing and placing into a tube furnace. For crystal growth, the temperature was ramped from room-temperature to 973 K, which was held constant for 14 days by PID control. After that, the furnace was naturally cooled to room temperature.

### Tunneling junction fabrications
CCPS MTJs were fabricated in an argon-filled glove box. Different thicknesses of CCPS flakes were exfoliated from single-crystal seeds, identified by optical contrast and further cross checked by atomic force microscopy (Park system NX10) and Raman spectroscopy. To obtain a MTJ shown in Fig. 1e, thin-layer CCPS flakes were exfoliated on $SiO_2$/Si substrates, followed by two PC/PDMS based dry-transfer steps[58] to prepare the top and bottom few-layer graphene (FLG) electrodes. In the final step, the MTJ is covered by a thin-film flake of hexagonal boron nitride to protect the sample from ambient exposure. Top and bottom FLG electrodes were contacted by 5/50 nm Cr/Au electrode using standard electron beam lithography (EBL) technique followed by thermal evaporation.

### Electrical measurements
Electrical measurements were performed in an Oxford-14 T cryostat. The system is equipped with a sample rotator for applying magnetic field either in-plane or out-of-plane. The $I-V$ curve were measured using a Keithley 2400 and zero-bias-resistance were measure by using standard lock-in method (with excitation frequency < 20 Hz).

### SHG measurements
The SHG measurements are conducted in a liquid nitrogen cooled optical cryostat with a femtosecond laser (Rainbow OEM, NPI Lasers). The wavelength is centered at either 1064 nm or 1560 nm, both yielding consistent SHG results. The laser beam is focused on the samples with a near-infrared radiation (NIR) objective (MplanApo NIR 50x, OptoSigma) and collected in a reflective geometry, wherein the SHG signal is separated with a dichroic mirror. The signal is detected with a fiber coupled spectrometer (UHTS 600 VIS, Witec). For polarization-dependent SHG measurements, the polarization angle between the incident laser and the SHG signal is set by two polarizers, and the in-plane polarization direction is rotated by a dual band half-wave plate before the optical objective. The optic setup is schematically illustrated in Supplementary Fig. 14d. The signals are confirmed to be originating in the CCPS samples by comparing with the reference spectra from BN, FLG electrodes and $SiO_2$ areas on the substrates.

### DFT calculations
The first-principles DFT calculations were performed using the Vienna ab initio simulation package (VASP) with the choice of projector augmented waves (PAW) basis set[59–61]. The Perdew-Burke-Ernzerh (PBE) functional has been employed to treat the exchange and correlation functional[62]. The energy cut off for the plane-wave basis was set to 600 eV. All the structures were fully relaxed until the force on every atom and energy were converged to 0.01 e Å$^{-1}$ and $1 \times 10^{-7}$ eV, respectively. A $\Gamma$-centred Monkhorst-Pack $12 \times 8 \times 1$ k-mesh was used for k-point sampling[63]. To model the 2D films, a vacuum layer of at least 20 Å was included in the $c$-axis direction. The Hubbard correction of $U = 3$ eV is applied to the $3d$ orbitals of Cr[64]. The vdW corrections were included by the DFT-D3 method[65]. The kinetic pathways were calculated by the climbing image nudged elastic band (CINEB) method[66].

### Layer-dependent CAFM saturation field $H_{sat}$
For in-plane $H_\parallel$, the magnetization of each ML ($M_s$) lies within the vdW plane, which can be described by a 1D spin-chain model[24]. By labeling the magnetization of the $i$th layer as $\mathbf{M}_i = M_s(\cos\theta_i, \sin\theta_i, 0)$, where $\theta_i$ is

the in-plane CAFM canting angle, the magnetic coupling energy of an N-layer CCPS is:

$$U_N(\theta_1,\ldots,\theta_N;H) = J\sum_{i=1}^{N-1}\cos(\theta_{i+1}-\theta_i) - \mu_0 M_s H\sum_{i=1}^{N}\cos(\theta_i). \quad (1)$$

The energy favorable CAFM state for a given $H$ can be obtained by solving the lowest-energy solutions of $U_N$,

$$\frac{\partial U_N}{\partial \theta_i} = 0 \quad \forall i = 1,\ldots,N. \quad (2)$$

For a solution set of $\{\theta_i^s = 1,\ldots,N\}$ satisfying the energy minimum requirement of $U_N$, the matrix of the second derivatives of $U_N$ must have all positive eigenvalues:

$$\det(A^s) > 0 \quad \text{with} \quad A_{ij}^s = \frac{\partial^2 U_N}{\partial\theta_i\partial\theta_j}\bigg|_{\theta_i=\theta_i^s}. \quad (3)$$

For the saturation FM′ state with $\theta_1 = \theta_2 = \ldots = \theta_N = 0$, the corresponding matrix $A^{FM'}$ has a much simplified tridiagonal form of,

$$A^{FM'} = \begin{pmatrix} H-\frac{H_J}{2} & \frac{H_J}{2} & & & & 0 \\ \frac{H_J}{2} & H-H_J & \frac{H_J}{2} & & & \\ & \frac{H_J}{2} & \cdots & \cdots & & \\ & & \cdots & \cdots & \frac{H_J}{2} & \\ & & & \frac{H_J}{2} & H-H_J & \frac{H_J}{2} \\ 0 & & & & \frac{H_J}{2} & H-\frac{H_J}{2} \end{pmatrix}, \quad (4)$$

where $H_J = 2J/(\mu_0 M_s)$ represents the energy induced by the interlayer exchange coupling $J$ ($J_\perp$). For $A^{FM'}$, the $(N-1)$ eigenvalues can be obtained by the standard approach for tridiagonal matrices, which reads:

$$\lambda_k^{FM'} = H - H_J + H_J\cos\left(\frac{k\pi}{N}\right) \quad \text{with} \quad k = 1,\ldots,N-1. \quad (5)$$

To get a positive matrix determinant, $H_{sat}$ must make the smallest eigenvalue for $k = N-1$ vanish:

$$H_{sat} - H_J + H_J\cos\left(\frac{N-1}{N}\pi\right) = 0, \quad (6)$$

which yield the analytical result of $H_{sat}$ by

$$H_{sat} = H_J + H_J\cos\left(\frac{\pi}{N}\right) = 2H_J\cos^2\left(\frac{\pi}{2N}\right). \quad (7)$$

Eq. (7) explains the layer-dependent CAFM saturation field $H_{sat}$, which increases substantially from 3.2 T for BL CCPS to 5.7 T for quintuple-layer CCPS. Using experimental data, the interlayer exchange coupling $J$ can also be extracted by fitting the layer-dependent $H_{sat}$ to Eq. (7).

## Data availability
The data that support the findings of this study are provided in the Source Data file. Source data are provided with this paper.

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

## Acknowledgements

This work is supported by the National Key R&D Program of the MOST of China (Grant Nos. 2023YFA1406302, and 2019YFA0308602), the National Science Foundation of China (Grant Nos. 12374194, 11574264, 12241401 and 12174336), and the Zhejiang Provincial Natural Science Foundation (D19A040001). Y.Z. acknowledges support from the Users with Excellence Project of Hefei Science Center CAS, 2021HSC-UE007. This work made use of the resources of the Center of Electron Microscopy of Zhejiang University.

## Author contributions

Y.Z. initiated and supervised the project. Q.F.H and M.J.Z. synthesized and characterized CCPS crystals. Q.F.H., M.J.Z. and Y.W. fabricated CCPS devices and carried out all the measurements. S.J.D. and C.H.J. did the ADF-STEM experiments. Y.Q.H., C.Q.H. and Y.H.L. did the DFT calculations. T.T. and K.W. synthesized the hBN crystals. X.F.X., J.B.Y., S.J.Y. and D.W.W. discussed the results and commented on the manuscript. Q.F.H., Y.Q.H., Y.W., L.J.L. and Y.Z. analysed the data and wrote the paper with inputs from all authors.

## Competing interests

The authors declare no competing interests.

## Additional information

[1]School of Physics, and State Key Laboratory of Silicon and Advanced Semiconductor Materials, Zhejiang University, Hangzhou 310027, China. [2]Zhejiang Province Key Laboratory of Quantum Technology and Device, School of Physics, Zhejiang University, Hangzhou 310027, China. [3]State Key Laboratory of Silicon and Advanced Semiconductor Materials, Zhejiang University, Hangzhou 310027, China. [4]State Key Laboratory of Extreme Photonics and Instrumentation, College of Optical Science and Engineering, Zhejiang University, Hangzhou 310027, China. [5]School of Physics Science and Engineering, Tongji University, Shanghai 200092, China. [6]State Key Laboratory for Mesoscopic Physics, School of Physics, Peking University, Beijing 100871, China. [7]School of Physics and Technology, Wuhan University, Wuhan 430072, China. [8]National Institute for Materials Science, 1-1 Namiki, Tsukuba 305-0044, Japan. [9]These authors contributed equally: Qifeng Hu, Yuqiang Huang, Yang Wang, Sujuan Ding, Minjie Zhang. ✉e-mail: lilinjun@zju.edu.cn; luyh@zju.edu.cn; chhjin@zju.edu.cn; phyzhengyi@zju.edu.cn

