## [Peer Review File · Nature Communications]

Ferrielectricity controlled widely-tunable magnetoelectric coupling in van der Waals multiferroicsREVIEWER COMMENTS

Reviewer #1 (Remarks to the Author):

The paper by Hu et al. deals with the ferrielectricity controlled widely-tunable magnetoelectric coupling in van der Waals multiferroics. Throughout the text, unfortunately, I did not see much novelty in this study by comparing it with the reported literature. The studies on magnetism and ferroelectricity in 2D vdW crystals are indeed an fascinating research topic with its multiple applications in devices. However, a ground-breaking finding is not presented. It is not recommended for publication in Nature Communications. Many questions are listed below.

1. The multiferroicity and magnetoelectric coupling of van der Waals CuCrP2S6 were widely reported. However, the authors seldom mention the previous literatures on van der Waals CuCrP2S6. I am doubt why authors ignore the recent important progress on van der Waals CuCrP2S6, see Nat Commun 14, 840 (2023); Nat Commun 7, 12357 (2016). ; Nanoscale 11, 5163–5170 (2019); Adv. Mater. Inter. 9, 2101769 (2022); Adv. Electron. Mater. 8, 2101072 (2022); Adv. Funct. Mater. 32, 2204214 (2022). Given the van der Waals CuCrP2S6 used here, it's no wonder to see the multiferroicity and tunable magnetoelectric coupling effects.
2. How about the quality and purity of samples? How about the phases? Please provide XRD results.
3. The authors report and highlight the unconventional MEC mechanism interlocked with heterogeneous ferrielectric (FiE) transitions. Here the “heterogeneous (FiE) transitions” are predicted from DFT calculations, valid experimental proofs are needed.
4. Spin-orbit coupling effects are critical for the magnetism and multiferroicity. Please perform DFT calculations with considering spin-orbit coupling effects.
5. How about the lattice dynamic stability of the modeled structure of CCPS in DFT calculations?
6. Authors ascribe the reversal behavior in $R \pm 0$ to the inherent AFE phase of CCPS, one open question remains that whether graphene layer contributes to the multiferroicity or magnetoelectric coupling results. It is widely reported that interfacial ferroelectricity could emerge by stacking non-polar materials in a symmetry-breaking way, see Nature 560, 336 (2018); Science 372, 1458–1462 (2021); Science 372, 1462–1466 (2021). First principles calculations demonstrate an enhanced magnetoelectric effect in the thin graphone/ferroelectric layers (Sci Rep 8, 12448 (2018) and graphene/In2Se3 heterostructure (ACS Appl. Mater. Interfaces 2021, 13, 2, 3033–3039)

Reviewer #2 (Remarks to the Author):

In this manuscript entitled “ferrielectricity controlled widely-tunable magnetoelectric coupling in van der Waals multiferroics”, Hu et al reported an electric field tunable magneto-electric coupling in two-dimensional CuCrP2S6. The magnetism in CuCrP2S6 can be controlled by external electric field. It is quite interesting and may provide a new route for electronically controlled spintronics. However, the manuscript cannot be accepted unless the following issues should be well addressed.

1. The significance, as well as the potential application, of this work was not clearly stated. It

is mentioned in the Abstract that “we show that an unconventional MEC mechanism interlocked...”. What does the “unconventional” mean? What’s the difference between different MECs? What does “interlocked” mean? It should be clearly explained.

2. More evidence to support the ferrielectricity of CuCrP2S6 should be provided. FiE state has the highest energy as shown in Figure 4b, and the energy of meta-AFE1 is lower than that of FiE. Why does the FiE phase emerge instead of the low-energy meta-AFE1 under electric field? How to confirm this phase is FiE, but not others?

3. How does the applied voltage affect the ferrielectricity of CuCrP2S6? If the ferrielectricity correlates with the rearrangement of Cu ions, the movement of Cu ions should be experimentally verified. How about the bias effect on the ferrielectricity of CuCrP2S6 with different layers, such as monolayer, bilayer, few-layer and multilayer? In Figure 3, only the 8L sample was studied. Do samples with different layers show the same behavior?

4. What is the mechanism of MEC? How does the Cu ions movement lead to the change of the valence state of Cr ions and thus the magnetism? Experiment evidence should be provided. External electric field, not the ferroelectricity, can also modulate the magnetism as shown in Fe3GeTe2 (Ref 10, 15). What’s the difference between those two kinds of modulation? How to obtain the 50% reduction in the interlayer magnetic coupling? Does the in-plane magnetic field direction influence the tunneling behavior?

5. Why are the work function of the top and bottom graphene electrodes different? How many layers of the graphene electrodes was used? Is the work function dependent on the layer number? Does the interface such as trapped impurities lead to the asymmetric tunneling current?

6. The manuscript should be improved. In Figure 2b, what do z , E , φ_P and φ_{AP} stand for? Which parts denote electrodes and CuCrP2S6? Does the spin configuration in the high R and low R state show the same direction? What does the inset in Figure 4b represent? The calculation detail of the depolarization field should be provided. How does the FiE transition minimize the depolarization effect (page 2)? “2D limit in vdW monolayers of CuCrP2S6 with inherent ferromagnetism and antiferroelectricity.” (page 2) will confuse the readers, since no monolayer was studied in the manuscript. There was no gate electrode used. Therefore, gate-tunable or similar statement would not be used.

7. Please carefully check the manuscript. It seems that the blue circle in the top middle panel of Figure 4a was missing. 337 meV was missing too. There is no meta-AFE2 states (page 8) in BL CuCrP2S6, and there are only meta-AFE1 and meta-AFE3 in Figure 4a. “along the H direction to from the canted AFM (CAFM) state” (page 4) should be “along the H direction from the canted AFM (CAFM) state”.

8. Some related works should be cited, such as Nanoscale 2019, 11, 5163, Phys. Rev. Mater. 2023, 7, 033402.

Reviewer #3 (Remarks to the Author):

The authors have reported on the tunneling magnetoresistance in the MTJs with graphene electrodes and a few-layer multiferroic CuCrP2S6 insulating barrier. The observed magnetic and electric tunability is both interesting and potentially important for understanding spin-dependent tunneling behaviors, as well as possible magnetoelectric coupling in 2D systems. However, there are several issues that need clarification before considering its publication.

1. The authors have determined the magnetic phase transition of bilayer MTJs solely from the temperature-dependent tunneling conductance data. Low-dimensional systems typically exhibit lower magnetic ordering temperatures than their bulk counterparts. Authors are

encouraged to provide additional (and possibly more direct) experimental evidence to support the magnetic phase transitions and their correlation with the tunneling conductance. It is recommended to include data depicting the thickness-dependent tunneling conductance vs. temperature.

2. The authors have also claimed the existence of a metamagnetic transition based solely on the TMR experiment, which might not be entirely convincing. To enhance the credibility of this claim, it is necessary to include magnetic data from either bulk or bilayer systems. Additionally, it would be beneficial to address questions about in-plane anisotropy. How does the magnetic transition differ when an in-plane H-field is applied perpendicular to or parallel with the spin direction?

3. The manuscript should provide a clear definition of TMR. In Figure 2(c), the TMR in the paramagnetic state is shown to be positive and increases with increasing H-field. Does this imply that the resistance is higher under high H-field conditions in the PM state? Is this behavior in line with the suggested model? In addition, TMR in the ordered state, as shown in Figure 3(a), increases as the H-field is raised. This seems to contradict the data in Figure 2(c), where TMR decreases until the field reaches 3.5 T in the ordered state. These discrepancies necessitate a thorough discussion.

4. It would be beneficial if the authors could include the number of layers for each figure, such as "N=2, N=5, N=10". Additionally, a comprehensive description of Figure 2(b) is necessary. The spin filter mechanism is difficult to comprehend from the figure alone, and a more detailed explanation of this mechanism is required in the manuscript.

5. On page 5, the term " V_b " is utilized without adequate explanation. Is it referring to the gate voltage? If so, where is this gate voltage applied?

6. In Figure 3(c), the tunneling current in the CAFM phase (at $H \approx 7$ T, $V_b = 1.6$ V) appears to have a higher magnitude (~ -1.2 μ A) than that ($\sim +0.9$ μ A) in the FM phase (at $H \approx 7$ T, $V_b = -2$ V). In other words, the CAFM phase under the strong H-field exhibits higher conductance than the FM phase under the same H-field. This behavior appears inconsistent with the proposed model, which would suggest that the CAFM phase should exhibit lower conductance compared to the FM phase. This discrepancy requires in-depth discussion.

7. In Figure 3(d), a negative V_b yields a negative I_t and vice versa. However, in Figure 3(c) at $H = 0$ T, a negative V_b (-2 V) yields a positive I_t and vice versa. Once again, this inconsistency necessitates proper discussion.

8. On page 7, the authors assert, "As a direct consequence of ultra-gate tunable ΔE_{MEC} , the E-enforced heterogenous FiE state has drastically reduced J_{\perp} , which changes by nearly 50 % with a moderate E of 0.3 V/nm as shown in Fig. 3." However, this description does not align with Figure 3, as no " ΔE_{MEC} " is provided. Moreover, the data exhibiting "the E-field dependent change of J_{\perp} value" is not apparent in any of the figures. These data are crucial for drawing conclusions and should be addressed.

9. On page 8, the authors mention, "The striking energy... the meta-AFE1 and meta-AFE2 states of BL CCPS...". However, as demonstrated in Figure 4 (the first row), there is no meta-AFE2 state in BL. Shouldn't it be meta-AFE3 state instead? The terms "meta-AFE2,3" need to be precisely defined in the manuscript.

10. On page 3, the authors declare, "Here, we reported ... tunable 2D MEC... fundamentally different from the bulk counterparts." However, recent researches have observed E-field induced local polarization and significant magnetoelectric coupling in bulk samples (Adv. Funct. Mater. 2022, 32, 2204214 & Adv. Electron. Mater. 2022, 2101072). The authors must carefully review recent progress in the bulk system to ensure accurate statements.

REVIEWER COMMENTS

Reviewer #1 (Remarks to the Author):

Comment 1-1: *“The paper by Hu et al. deals with the ferrielectricity controlled widely-tunable magnetoelectric coupling in van der Waals multiferroics. Throughout the text, unfortunately, I did not see much novelty in this study by comparing it with the reported literature. The studies on magnetism and ferroelectricity in 2D vdW crystals are indeed an fascinating research topic with its multiple applications in devices. However, a ground-breaking finding is not presented. It is not recommended for publication in Nature Communications. Many questions are listed below.”*

Reply 1-1:

We thank the Reviewer for critically summarizing the topic of our work. The novelty of this work, as pointed out by the Reviewer, is **the discovery of an unconventional ferrielectricity controlled magnetoelectric coupling (MEC) mechanism in vdW multiferroic CuCrP₂S₆**. The concept of vdW multiferroics is currently heavily explored by the interdisciplinary 2D research community, with a special focus on unveiling unconventional MEC effects in the 2D limit. In bulk multiferroics, MEC is either extremely weak with nearly independent magnetism and ferroelectricity (type-I multiferroic), or requires the Dzyaloshinskii-Moriya interaction (type-II, spin-driven multiferroic). In the 2D limit, unconventional MEC mechanisms are expected to emerge due to extra tunability introduced by external modulation forces, such as electric field and strain, and by interface engineering, i.e. the two-phase approach [see Nature 442, 759 (2006)]. With the unique vdW layered structure, emergent 2D materials provide a fascinating playground for exploring unconventional MEC mechanisms in the 2D limit. Indeed, utilizing an external electric field of 0.3 V/nm, **we demonstrate an unconventional MEC mechanism driven by layer-dependent ferrielectric transitions, which allows interlayer magnetic coupling energy to be efficiently tuned by nearly 50% in thin-layer CuCrP₂S₆**. To the best of our knowledge, this is the very first report of novel MEC mechanisms in vdW 2D materials, and surely, we are expecting to see more exotic MEC phenomena in the 2D monolayer limit.

It should also be emphasized that 2D material based multiferroic and MEC studies are rather new research topics, with the first experimentally claimed single-layer multiferroic very recently in NiI₂ [see Nature 602, 601 (2022)]. Although the existence of ferroelectricity in NiI₂ is controversial, it shows the great promises of vdW multiferroics for realizing efficient MEC, considering that it is not rare for a 2D vdW system to host multiple ferroic orders.

In response to the Reviewer comments and various professional suggestions, we have updated and revised the paper accordingly, which are summarized below in a point-to-point fashion.

Comment 1-2: *“The multiferroicity and magnetoelectric coupling of van der Waals CuCrP₂S₆ were widely reported. However, the authors seldom mention the previous literatures on van der Waals CuCrP₂S₆. I am doubt why authors ignore the recent*

important progress on van der Waals CuCrP₂S₆, see Nat Commun 14, 840 (2023); Nat Commun 7, 12357 (2016). ; Nanoscale 11, 5163–5170 (2019); Adv. Mater. Inter. 9, 2101769 (2022); Adv. Electron. Mater. 8, 2101072 (2022); Adv. Funct. Mater. 32, 2204214 (2022). Given the van der Waals CuCrP₂S₆ used here, it's no wonder to see the multiferroicity and tunable magnetoelectric coupling effects."

Reply 1-2: We appreciate these critical comments on our work from the Reviewer, and we agree that these excellent works on **bulk CuCrP₂S₆** (CCPS) should be acknowledged. As a response, we have properly cited these literatures in this revised version.

The multiferroicity in bulk CuCrP₂S₆ crystals, i.e. the coexistence of interlayer anti ferromagnetism and in-plane stripe antiferroelectricity, is well established. However, it is widely known that for most of 2D materials, **the bulk properties can be fundamentally different from thin-layer flakes**, due to the weakening of interlayer interaction, quantum confinement and symmetry changes. Equally important, **the multiferroic coupling mechanism of bulk CuCrP₂S₆ is not clear at the moment**, for example Youfang Lai et al reported that it is a type-I multiferroic, while Chang Bae Park et al claimed a type-II mechanism. It should also be noticed that **a multiferroic coexistence does not guarantee efficient or tunable magnetoelectric coupling in bulk or in the 2D limit**, which should only be validated by systematic experiments.

Indeed, dimension reduction may induce profound changes in the physical properties of CuCrP₂S₆, e.g. Youfang Lai et al reported the ferroelectric poling of CuCrP₂S₆ flakes with ~13 nm thicknesses. However, for CuCrP₂S₆, **the research of magnetic phases in the 2D limit and magnetoelectric coupling in few-layer CCPS have not been demonstrated yet**. In this study, **we report a completely different type of multiferroic coupling mechanism driven by layer-dependent ferrielectric transitions in thin-layer CCPS**, which are efficiently controlled by an external electric field and **should not be categorized into neither type-I nor type-II multiferroic**. To the best of our knowledge, this is the very first report of novel MEC mechanisms in vdW 2D materials, which will meet the scope and publication criteria of Nature Communications.

Related changes are:

- I. Main Text, Page 13, New References [44,46-49]
- II. Main Text, Page 2, Paragraph 1, Line 5-6, "...in paradigmatic bulk systems, viz. type-I multiferroics with nearly independent magnetism and ferroelectricity and type-II multiferroic driven by the Dzyaloshinskii-Moriya interaction,..."

Comment 1-3: *"How about the quality and purity of samples? How about the phases? Please provide XRD results."*

Reply 1-3: We thank the Reviewer for this advice. We have used high quality single crystals of CuCrP₂S₆ for the few-layer and multilayer device fabrications. Attached below shows the typical XRD results of single crystal CCPS, which are collected for each batch of samples after the self-flux syntheses (see Methods in the main text). The high quality of our bulk crystals is evident by the XRD peaks of (002), (004) and (008) corresponding to the van der Waals planes. An optical image of CCPS single crystals, showing shining crystalline surfaces, is also provided as the inset for the reference of the Reviewer and the potential readers.

In response, we have added the related Figure into the Supplementary Information as new Figure S1.

Comment 1-4. The authors report and highlight the unconventional MEC mechanism interlocked with heterogeneous ferroelectric (FiE) transitions. Here the “heterogeneous (FiE) transitions” are predicted from DFT calculations, valid experimental proofs are needed.

Reply 1-4: The Reviewer’s insightful comment is deeply appreciated. From physics point of view, **the unconventional MEC coupling mechanism in thin-layer CCPS flakes is correlated to the copper ion movement in response to external electric field, which is not only based on DFT calculations but also confirmed by temperature- and bias-dependent second harmonic generation (SHG) experiments.** The SHG technique is an established and ultra-sensitive tool for probing broken inversion symmetry. By fabricating multilayer CCPS devices with the heterostructures of graphene/BN/CCPS/BN/graphene, we utilize SHG to directly confirm the Cu⁺ movement and the heterogenous FiE phase controlled by electric field.

In response, we have added the SHG results into the main text and the Supplementary Materials as new Figure 4c, 4d and Figure S14. Related changes are the following:

- I. Main Text, Page 22, Figure 4 Captions, Line 12-14, we have added the sentences: “c, SHG intensity vs T, showing the AFE transition and the frozen of Cu⁺ ions at around 140 K. d, V_b-tunable SHG intensity, showing a signal change over a factor of 3 from -1 V to 1 V.”
- II. Main Text, Page 7, Paragraph 2, Line 8-20, “The formation of an E-dependent FiE state is confirmed by second harmonic generation (SHG) experiments, with tunable V_b applied to the heterostructures of FLG/BN/CCPS/BN/FLG. As shown in Fig. 4c, by cooling down a 6L device, the T-dependent SHG intensity decreases

monotonically and eventually stabilizes at about 140 K, which manifests the AFE transition of thin-layer CCPS when Cu^+ ions lock into the stripe-AFE positions. At 77 K, by ramping the V_b from -1 V to 1 V, the SHG intensity changes by more than three times. Similar results are well reproduced on multiple devices, and also repeated on an FLG/CCPS/SiO₂/Si structure, in which the heavily doped silicon substrate was used as the bias electrode. The V_b -dependent SHG intensity reflects continuous inversion symmetry breaking induced by \mathbf{E} . As illustrated in the insets of Fig.4d, for the stripe-AFE ground, there exists an inversion center between two neighboring monolayers, while the V_b -driven heterogeneous FIE state has reduced inversion symmetry due to layer-dependent rearrangements of anti-parallel Cu^+ ions within the FM vdW cages.”

Figure S14. **SHG measurement of CCPS.** **a**, Measured SHG spectrum of an octuple-layer CCPS flake upon excitation of 1064 nm laser. **b**, A septuple-layer CCPS device with double BN encapsulations and top and bottom graphene electrodes for applying electric field.

Figure 4c and 4d. **SHG measurements of CCPS.** **c**, SHG intensity as a function of temperature, showing the AFE transition and the frozen of Cu^+ ions at around 140 K. **d**, CCPS SHG intensity versus V_b from -1 V to 1 V, causing the SHG intensity to change by a factor over 3.

Comment 1-5. Spin-orbit coupling effects are critical for the magnetism and multiferroicity. Please perform DFT calculations with considering spin-orbit coupling effects.

Reply 1-5: By considering SOC effects, we calculate ΔE_{MEC} a function of Cu^+ ion movements in the FM vdW cages. As shown in the Figure attached below, the results are essentially the same as the original one without SOC. **This is not so surprising since the orbital quenching effect is still predominant in CCPS, due to the octahedron crystal field environments for Cr ions.**

In response to the Reviewer's question, we have added the figure into the Supplementary Information as Figure S15.

Comment 1-6. *How about the lattice dynamic stability of the modeled structure of CCPS in DFT calculations?*

Reply 1-6: We have indeed verified the lattice dynamic stability. Related results have been published in [Chin. Phys. Lett. 38, 077501 (2021)], by one of our previous group member Dr. Chenqiang Hua.

In response to the Reviewer's question, we have added new Reference [50] in the main text.

Comment 1-7. Authors ascribe the reversal behavior in $R_{\pm 0}$ to the inherent AFE phase of CCPS, one open question remains that whether graphene layer contributes to the multiferroicity or magnetoelectric coupling results. It is widely reported that interfacial ferroelectricity could emerge by stacking non-polar materials in a symmetry-breaking way, see Nature 560, 336 (2018); Science 372, 1458–1462 (2021); Science 372, 1462–1466 (2021).

First principles calculations demonstrate an enhanced magnetoelectric effect in the thin graphone/ferroelectric layers (Sci Rep 8, 12448 (2018) and graphene/ In_2Se_3 heterostructure (ACS Appl. Mater. Interfaces 2021, 13, 2, 3033–3039)

Reply 1-7: We thank the Reviewer for raising this discussion. There are indeed many fascinating experimental reports on unconventional ferroelectricity in the 2D limit, such as 2D ferroelectricity in semimetal two and three-layer WTe_2 [Nature 560, 336 (2018)], and parallel stacking-induced ferroelectricity in bilayer boron nitrides [Science 372, 1458–1462

(2021); Science 372, 1462–1466 (2021)], in both works graphene electrodes are used as a sensor for probing the out-of-plane polarization. However, **by creating a graphene/2D ferroelectric heterostructure does not guarantee an interfacial magnetoelectric effect.** For example, the theoretical proposals of [Sci. Rep. 8, 12448 (2018)] and [ACS Appl. Mater. Interfaces 13, 3033–3039 (2021)] are basically the two-phase approach combining a ferromagnet and a ferroelectric [see Nature 442, 759 (2006)], in which graphene is either hydrogenated or decorated with transition metal atoms.

We can also exclude the possible MEC origin of thin-flake CCPS in ferroelectric switching, which manifests in the I - V characteristics as a hysteresis behavior. In our experiments, hysteresis in the I - V curves is completely absent, and the deduced $R_{\pm}(0)$ are highly reproducible and represent the dichotomic tunnelling resistance under opposite V_b with a zero magnetic field. From the energy point of view, the absence of hysteresis in our I - V results is reasonable since the energy barrier of the antiferroelectric-ferroelectric (AFE-FE) transition is too steep to be overcome by a realistic gate voltage before introducing electrical breakdown. Furthermore, we have pushed the thicknesses of I - V measurements to one and two monolayers, which consistently show no hysteretic behaviors (the data are attached below for the reference of the Reviewer).

In response to the Reviewer's question, we have added these related references into the revised manuscript. See Main Text, Page 11 and Page 12 for new References [18-19, 33-34].

Reviewer #2 (Remarks to the Author):

Comment 2-1: In this manuscript entitled “ferrielectricity controlled widely-tunable magnetoelectric coupling in van der Waals multiferroics”, Hu et al reported an electric field tunable magneto-electric coupling in two-dimensional CuCrP2S6. The magnetism in CuCrP2S6 can be controlled by external electric field. It is quite interesting and may provide a new route for electronically controlled spintronics. However, the manuscript cannot be accepted unless the following issues should be well addressed.

Reply 2-1: We gratefully appreciate the Reviewer for the expert reviewing and positive evaluation of our work. In particularity, we thank the Reviewer for comments that “*quite interesting and may provide a new route for electronically controlled spintronics*”. We have addressed the Reviewer’s concerns by revising the paper accordingly, fully taking his/her comments into considerations. The changes are summarized below in a point-to-point fashion.

Comment 2-2: The significance, as well as the potential application, of this work was not clearly stated.

Reply 2-2: Pioneered in the 1950s, the research field of multiferroic study has long been bulk material based, and it has a major application-oriented theme of pursuing the mutual control of different *ferroic* orders, in particularly between ferromagnetism and ferroelectricity. However, such a **magnetoelectric type of multiferroics are rather rare in the bulk form, which are either type-I multiferroic with nearly independent magnetism and ferroelectricity, or type-II multiferroic driven by the Dzyaloshinskii-Moriya interaction**. The lacking of a single-phase multiferroic bulk system with efficient magnetoelectric coupling has pushed the field to seek alternative magnetoelectric materials with unconventional tunable parameters, such as **two-phase systems** consisting of multiple ferroic functional-layers, and novel MEC coupling mechanism via strain and interface engineering etc [see review in Nature 442, 759 (2006)].

In this general background, **we explore the feasibility to realize unconventional MEC coupling mechanisms in thin-layer vdW crystals**, which offer rich choices of ferroic atomic layers with single-crystal quality that can be further tuned by electric field, strain, interfaces and spin-lattice interactions. Although the electric field control of magnetism has been reported in several 2D crystals, the mutual control of magnetism and ferroelectricity in a vdW multiferroic material has not been realized yet. **Our study for the first time achieves the electric field control of magnetic phases in thin-layer CCPS devices**, in which the indirect magnetoelectric coupling is driven by heterogeneous ferrielectric (FiE) transitions of the intralayer stripe AFE phase. Our research showcases that CCPS is a fascinating 2D multiferroic material with great potential in device applications as well as on studying the multiple ferric order interaction in the 2D limit. It also provides an amazing example on how dimensionality control may induce profound changes in the physical properties of vdW crystals (we also noticed that the recent paper on [Nat. Commun. 14, 7304 (2023)] reported the observation of robust out-of-plane ferroelectricity in thin-film CCPS at room temperature).

The aforementioned background information on bulk multiferroic and MEC studies and the key findings of our paper on unconventional MEC coupling in thin-layer CCPS are well presented in the first two paragraphs of the main text. **However, we fully agree with the Referee that beyond the multiferroic and MAE topics, electronically controlled spintronics may also be an appealing direction for thin-layer CCPS, considering the highly efficient tuning of the canted AFM saturation energy by external electric field.**

In response to the Reviewer’s comment, we have added discussion on the perspective of electronically controlled spintronics based on thin-layer CCPS into the revised manuscript. See Main Text, Page 9, Paragraph 1, Line 5-8 “Equally important, the

multiferroicity in thin-layer CCPS opens the possibility for the investigation of 2D quantum phase transitions, and the fabrication of complex vdW heterostructure for the realization of two-phase multiferroic for novel electronically controlled spintronics and Josephson junctions.”

Comment 2-3: It is mentioned in the Abstract that “we show that an unconventional MEC mechanism interlocked...”. What does the “unconventional” mean? What’s the difference between different MECs? What does “interlocked” mean? It should be clearly explained.

Reply 2-3: We appreciate the Reviewer for raising these questions. As we have elaborated in Reply 2-2, the multiferroic and MEC studies have a long history dated back to the 1950s. For **bulk single-phase multiferroics**, the coupling between ferromagnetism and ferroelectricity can be categorized into **type-I multiferroic with independent magnetism and ferroelectricity**, and **type-II multiferroic driven by the Dzyaloshinskii-Moriya interaction**.

The unconventional MEC coupling mechanism in thin-layer CCPS is driven by layer-dependent ferrielectric (FiE) transitions via copper ion movement within the ferromagnetic CrS₆-P₂S₆ cages, which should not be categorized into neither type-I nor type-II multiferroic. The physical origin of this indirect AFE-FM magnetoelectric coupling mechanism in thin-layer CCPS not only defines the uniqueness (unconventional nature) but also means it is interlocked with the heterogeneous FiE phase transitions.

In response to the Reviewer’s question, we have revised the main text accordingly to include the bulk multiferroic classifications. See: Main Text, Page 2, Paragraph 1, Line 5-9, “...in paradigmatic bulk systems, viz. type-I multiferroic with nearly independent magnetism and ferroelectricity and type-II multiferroic driven by the Dzyaloshinskii-Moriya interaction,...”

Comment 2-4. More evidence to support the ferrielectricity of CuCrP₂S₆ should be provided.

Reply 2-4: The **intralayer stripe antiferroelectric (AFE) phase** in CCPS bulk crystals is well understood (see Ref. [40-51] in the main text). Here, **for thin-layer CCPS, we use second harmonic generation (SHG) to experimentally confirm the AFE phase in the 2D limit**, and demonstrate that copper ions can indeed move within the ferromagnetic CrS₆-P₂S₆ cages under external electric field. **Related changes are summarized below (see also Reply 1-3):**

- I. We have added the SHG results into the main text as Figure 4c and 4d, and the Supplementary information as new Figure S14.
- II. Main Text, Page 22, Figure 4 Captions, Line 12-14, we have added the sentences: “**c**, SHG intensity vs T, showing the AFE transition and the frozen of Cu⁺ ions at around 140 K. **d**, V_b-tunable SHG intensity, showing a signal change over a factor of 3 from -1 V to 1 V.”
- III. Main Text, Page 7, Paragraph 2, Line 8-20, “The formation of an **E**-dependent FiE state is confirmed by second harmonic generation (SHG) experiments, with tunable V_b applied to the heterostructures of FLG/BN/CCPS/BN/FLG. As shown in Fig. 4c, by cooling down a 6L device, the T-dependent SHG intensity decreases

monotonically and eventually stabilizes at about 140 K, which manifests the AFE transition of thin-layer CCPS when Cu^+ ions lock into the stripe-AFE positions. At 77 K, by ramping the V_b from -1 V to 1 V, the SHG intensity changes by more than three times. Similar results are well reproduced on multiple devices, and also repeated on an FLG/CCPS/ SiO_2/Si structure, in which the heavily doped silicon substrate was used as the bias electrode. The V_b -dependent SHG intensity reflects continuous inversion symmetry breaking induced by \mathbf{E} . As illustrated in the insets of Fig.4d, for the stripe-AFE ground, there exists an inversion center between two neighboring monolayers, while the V_b -driven heterogeneous FiE state has reduced inversion symmetry due to layer-dependent rearrangements of anti-parallel Cu^+ ions within the FM vdW cages.”

Figure S14. **SHG measurement of CCPS.** **a**, Measured SHG spectrum of an octuple-layer CCPS flake upon excitation of 1064 nm laser. **b**, A septuple-layer CCPS device with double BN encapsulations and top and bottom graphene electrodes for applying electric field.

Figure 4c and 4d. **SHG measurements of CCPS.** **c**, SHG intensity as a function of temperature, showing the AFE transition and the frozen of Cu^+ ions at around 140 K. **d**, CCPS SHG intensity versus V_b from -1 V to 1V, causing the SHG intensity to change by a factor over 3.

Comment 2-5. *FiE state has the highest energy as shown in Figure 4b, and the energy of meta-AFE1 is lower than that of FiE. Why does the FiE phase emerge instead of the low-energy meta-AFE1 under electric field? How to confirm this phase is FiE, but not others?*

Reply 2-5: We thank the Reviewer for raising the question. The purpose of Figure 4b is to show that all **metastable antiferroelectric (mAFE) states are not energy accessible by experimental electric fields (E), starting from an intralayer stripe antiferroelectric (stripe-AFE) ground state.** These meta-AFE states are differentiated from the **ferrielectric (FiE) state** by Cu^+ ion movement across the energy barrier maximum located at the middle point of the FM $\text{CrS}_6\text{-P}_2\text{S}_6$ cage, i.e. the Cr atomic plane.

In brief, the potential energy vs Cu^+ ion displacement diagram is plotted by calculating the energy cost of flipping a minimal AFE stripe domain for thin-layer CCPS (results of bilayer and trilayer are summarized in Fig. S7 in the Supplementary Information which is also attached below). As illustrated in the figure below, there are multiple ways to flip one anti-parallel stripe domain, and the resulting mAFE states are indexed by layer number-prefix and Roman numeral-suffix (e.g. 2L-mAFE-I stands for the flipping of anti-parallel Cu^+ ions in the bottom ML), while the **fully polarized ferroelectric (FE) state** corresponds to the flipping of all anti-parallel stripe domains. For bilayer CCPS, we can see that even with an unrealistic $\mathbf{E}=2$ V/nm, the energy of the lowest energy 2L-mAFE-I state is still far above the stripe AFE ground state.

Although all the mAFE states are not polarizable by \mathbf{E} , thin-layer CCPS spontaneously respond to external electric fields by moving anti-parallel copper ions toward the Cr atomic plane within the FM $\text{CrS}_6\text{-P}_2\text{S}_6$ cages. Due to the unique vdW screening effect between neighboring AFE monolayers, the response of anti-parallel Cu^+ ions in different monolayers are not equivalent, producing a layer-dependent heterogenous FiE phase. **Such a formation of heterogenous FiE phase is not only based on DFT calculations but also confirmed by temperature- and bias-dependent second harmonic generation (SHG) experiments,** as elaborated in Reply 2-4 (see also Reply 1-3).

In response to the question, we have revised the article accordingly. Related changes are the following:

- I. Main Text, Page 6, Paragraph 2, Line 15; Page 6, Paragraph 3, Line 9; Page 7, Paragraph 1, Line 1-2; Page 9, Paragraph 1, Line 1; and Page 22, Figure 4 Labels and Captions, different mAFE states are properly named by layer number-prefix and Roman numeral-suffix, such as “2L-mAFE-I, 2L-mAFE-II” and “3L-mAFE-I, 3L-mAFE-II, 3L-mAFE-III”.
- II. The lattice model representation of quadruple-layer CCPS is moved into Supplementary Information as Figure S8, in which different mAFE states are named by “4L-mAFE-I, 4L-mAFE-II, 4L-mAFE-III”.
- III. We have added the SHG results into the main text as Figure 4c and 4d, and the Supplementary information as new Figure S14.
- IV. Main Text, Page 22, Figure 4 Captions, Line 12-14, we have added the sentences: “**c**, SHG intensity vs T, showing the AFE transition and the frozen of Cu^+ ions at around 140 K. **d**, V_b -tunable SHG intensity, showing a signal change over a factor of 3 from -1 V to 1 V.”
- V. Main Text, Page 7, Paragraph 2, Line 8-20, “The formation of an \mathbf{E} -dependent FiE state is confirmed by second harmonic generation (SHG) experiments, with tunable V_b applied to the heterostructures of FLG/BN/CCPS/BN/FLG. As shown in Fig. 4c, by cooling down a 6L device,

the T-dependent SHG intensity decreases monotonically and eventually stabilizes at about 140 K, which manifests the AFE transition of thin-layer CCPS when Cu^+ ions lock into the stripe-AFE positions. At 77 K, by ramping the V_b from -1 V to 1 V, the SHG intensity changes by more than three times. Similar results are well reproduced on multiple devices, and also repeated on an FLG/CCPS/ SiO_2/Si structure, in which the heavily doped silicon substate was used as the bias electrode. The V_b -dependent SHG intensity reflects continuous inversion symmetry breaking induced by \mathbf{E} . As illustrated in the insets of Fig.4d, for the stripe-AFE ground, there exists an inversion center between two neighboring monolayers, while the V_b -driven heterogeneous FiE state has reduced inversion symmetry due to layer-dependent rearrangements of anti-parallel Cu^+ ions within the FM vdW cages.”

Comment 2-6. How does the applied voltage affect the ferroelectricity of CuCrP2S6 ?

Reply 2-6: We thank the Reviewer's for raising this question. First, we want to emphasize that the ground state of thin-layer CCPS remains as the intralayer stripe-AFE ordering, which is confirmed by the temperature-dependent SHG experiments. Equally important, the ferroelectric state as well as the intermediate mAFE states are not energy accessible by realistic electric fields, as we have elaborated in Reply 2-5. Indeed, in our experiment results, all thin-layer devices do not show ferroelectric hysteresis loop. Instead, thin-layer CCPS spontaneously respond to external electric fields by moving anti-parallel copper ions

toward the Cr atomic plane within the FM $\text{CrS}_6\text{-P}_2\text{S}_6$ cages, leading to the formation of the **V_b -dependent heterogenous FiE phase.**

In brief, the voltage bias introduced FiE phase transitions can be summarized as follows using a 10L CCPS device as an example: for $V_b=0$ V, the device is under a built-in electric field which is attributed to the work function differences between two graphene electrodes; by applying a positive V_b to the bottom electrode, the positions of anti-parallel Cu^+ stripes in each monolayer rearrange by moving towards the Cr atomic plane, with distances Δd determined by the balance of the external electric field \mathbf{E} and the layer-dependent vdW depolarization field \mathbf{E}_{Dep} , i.e. $\mathbf{E} = \mathbf{E}_{\text{Dep}}$; when switching to a negative V_b , all the bottom Cu^+ stripes will move towards the balanced positions of the strip-AFE ground state, followed by the top Cu^+ stripes rearrangements towards the Cr atomic plane. The **anisotropic V_b -polarity dependent responses of the heterogenous FiE phase transitions**, which are confirmed by the SHG experiments, create two V_b -independent ΔE_{MEC} plateaus in the 2D tunneling conductance vs H and V_b data of Figure S6d, corresponding to the saturated FiE state and the stripe-AFE state respectively.

Comment 2-7. If the ferrielectricity correlates with the rearrangement of Cu ions, the movement of Cu ions should be experimentally verified.

Reply 2-7: We strongly agree with the Reviewer that the DFT calculations should be experimentally verified. Indeed, we have utilized temperature- and V_b -dependent second harmonic generation (SHG) measurements to experimentally prove that copper ions are continuously rearranging their positions within the FM $\text{CrS}_6\text{-P}_2\text{S}_6$ cages in response to an external bias voltage. The related results are summarized in Reply 2-4 (see also Reply 1-3).

Comment 2-8. How about the bias effect on the ferrielectricity of CuCrP_2S_6 with different layers, such as monolayer, bilayer, few-layer and multilayer? In Figure 3, only the 8L sample was studied. Do samples with different layers show the same behavior?

Reply 2-8: We thank the Reviewer for raising the intriguing question. For multilayer devices ($\geq 4\text{L}$), there are distinctive *even-odd* effects below 2 T, where odd-layer flakes exhibit a

field-induced perfect collinear AFM state. While for few-layer samples ($\leq 3L$), interfacial coupling effects between graphene electrodes and CCPS flakes are not negligible [see Nat. Nanotechnol. 17, 1272 (2022)], producing large background tunneling currents which predominate the canted-AFM transition induced changes in I_t . Even more technically challenging is that few-layer devices are rather vulnerable to electric field breakdowns (<0.2 V/nm), which make systematic study of the V_b -dependent FiE transitions infeasible. In contrast, for multilayer CCPS devices with minimized interfacial coupling effects, the breakdown electric field can be readily maximized to reach 1 V/nm, allowing us to fully unfold the anisotropic V_b -polarity dependent responses of the heterogenous FiE phase transitions.

In response to the Reviewer's question, we have added the V_b -tunable MEC data in a quintuple-layer MTJ. **Related changes are:**

- I. Normalized TMR vs H_{\parallel} curves and the 2D contour plot of the normalized TMR vs V_b and H_{\parallel} of MTJ 5L-S32 are added into the Supplementary Information as new Figure S16, and the result shows the same behavior with the MTJ 8L-S20 in the main text.

Comment 2-9. What is the mechanism of MEC? How does the Cu ions movement lead to the change of the valence state of Cr ions and thus the magnetism? Experiment evidence should be provided.

Reply 2-9: We thank the Reviewer for these professional comments. **The indirect MEC coupling mechanism in thin-layer CCPS is driven by layer-dependent FiE transitions via copper ion rearrangements within the FM $\text{CrS}_6\text{-P}_2\text{S}_6$ cages, which efficiently change the interlayer superexchange coupling energy by distorting the CrS_6 octahedrons and thus, weakening the S-anion exchange pathways.**

In response, we have revised the paper accordingly to include the aforementioned discussions on the physical origin of the MEC mechanism:

- I. We have added a schematic illustration of the interlayer superexchange controlled magnetoelectric coupling mechanism into the Main Text as new Figure 4e.
- II. Main text, Page 8, Paragraph 1, Line 4-15, “As illustrated in Fig. 4e, for thin-layer CCPS, the interlayer superexchange that determines J_{\perp} is intermediated by the interfacial majority spin p -orbitals of S atoms, which are antiparallel to the Cr magnetic moments due to the formation of Cr-S valence bonds. The stripe-AFE ground state corresponds to a relative symmetric CrS_6 octahedron with minimal lattice distortions (Fig. 4e and DFT calculations in SI Table S1). In response to external electric field, anti-parallel Cu^+ ions move within the FM vdW cages, enforcing substantially lattice distortions for the CrS_6 octahedrons by stretching the average Cr-S bond lengths. The resulting attenuation of the Cr-S bonding energy effectively decreases the net majority spins of S atoms, which changes from 0.135 electrons/S for $d=0 \text{ \AA}$ (the equilibrium stripe-AFE position) to 0.092 electrons/S for $\Delta d=0.44 \text{ \AA}$, and thus, efficiently lowers the interlayer AFM coupling.”
- III. Main text, Page 22, Figure 4 Captions, Line 14-15, “e, Illustration of the FiE-interlocked MEC mechanism in CCPS, in which lattice distortions introduced by Cu^+ ions displacements efficiently lower the interlayer superexchange interaction.”
- IV. We have added new Table S1 in the Supplementary Information to show Cu^+ ions movement introduced changes in effective Cr-S bond lengths.

	$\bar{x}(\text{\AA})$	$\overline{\Delta x}(\text{\AA})$	Cr-d(dn)	S-p(dn)	S-p(up)	S-p(up-dn)
AFE	2.4758	0.00443	3.385	2.680	2.815	0.135
0.22 \AA	2.4759	0.00509	3.396	2.681	2.798	0.117
0.44 \AA	2.476	0.00592	3.414	2.683	2.775	0.092
0.65 \AA	2.478	0.00879	3.421	2.684	2.758	0.074

Table S1. Calculated average Cr-S bond lengths and net majority spins of S-atoms within a CrS₆ octahedron as a function of anti-parallel Cu⁺ ion displacements within the FM CrS₆-P₂S₆ cages.

Comment 2-10. External electric field, not the ferroelectricity, can also modulate the magnetism as shown in Fe₃GeTe₂ (Ref 10, 15). What's the difference between those two kinds of modulation?

Reply 2-10: We thank the Reviewer for raising this question. **Fe₃GeTe₂ is an itinerant 2D ferromagnetic material, fundamentally different from insulating 2D multiferroic CCPS.** In the study of Fe₃GeTe₂ 2D magnetism [Nature 563, 94–99 (2018)], the modulation of Curie temperature up to room temperature is achieved by ultra-high electron doping via liquid ion gating. Such an approach is expected for itinerant ferromagnetic materials, in which **the Stoner criterion** determines the spontaneous ferromagnetic polarization by the density of states (DOS) at the Fermi level (other mechanism such as RKKY interactions may also contribute to the magnetism enhancement).

While for insulating thin-layer CCPS, electron doping is not a practical way to tune the magnetism as constrained by the large energy gap. Instead, the magnetism of thin-layer CCPS is modulated by **an indirect MEC coupling mechanism driven by layer-dependent FiE transitions via copper ion rearrangements within the FM CrS₆-P₂S₆ cages, which efficiently control the interlayer superexchange coupling energy by distorting the CrS₆ octahedrons** (see also Reply 2-9). Therefore, these two kinds of magnetism modulation mechanisms are rather different.

Comment 2-11. How to obtain the 50% reduction in the interlayer magnetic coupling?

Reply 2-11: **As we have elaborated in Reply 2-9 and Reply 2-10, the interlayer magnetic coupling is efficiently tuned by an indirect MEC coupling mechanism driven by heterogenous FiE transitions via copper ion rearrangements within the FM CrS₆-P₂S₆ cages, which effectively control the interlayer superexchange coupling energy by distorting the CrS₆ octahedrons.**

Quantitatively, the interlayer magnetic coupling of thin-layer CCPS can be modeled by an **1D spin chain model** [Nat. Nanotechnol. 14, 1116–1122 (2019)], which works well for **2D vdW magnets with intralayer ferromagnetism and interlayer antiferromagnetic coupling**. By minimizing the free energy of the 1D spin chain model, we can explicitly deduce the relation between the saturation magnetic field H_{sat} and the interlayer magnetic coupling parameter J_{\perp} :

$$H_{\text{sat}} = \frac{4J_{\perp}}{\mu_0 M_s} \cos^2 \frac{\pi}{2N},$$

in which M_s is the net magnetization of an individual monolayer and N is the layer number. **The linear relation between H_{sat} and J_{\perp} provides a viable way to experimentally determine the V_b -dependent interlayer magnetic coupling energy, since H_{sat} is a physical quantity measurable by the tunnelling spectroscopy.**

In the case of Figure 3a in the main text (MTJ 8L-S20), by analyzing the H_{sat} values for bias voltage 1.2 V and -1.2 V respectively, we determine a 50% reduction in H_{sat} by reversing the electric field, which is equivalent to a 50% reduction of the interlayer magnetic coupling strength J_{\perp} . Note that we have carried out the first-derivative and second-derivative analyses of the original TMR spectra presented in Figure 3a. Both methods consistently produce the same 50% reduction in H_{sat} , as summarized in Figure S5 which is also attached below for the convenience of the referees.

Figure S5. Analyses of the CAFM saturation field H_{sat} .

Comment 2-12. Does the in-plane magnetic field direction influence the tunneling behavior?

Reply 2-12: We thank the Reviewer for raising this question. From magnetism point of view, **the intralayer ferromagnetic ordering within a CCPS monolayer has an easy-plane magnetocrystalline anisotropy which lies in the van der Waals basal plane.** Just like the archetypical easy-plane vdW magnet CrCl_3 [Nat. Nanotechnol. 14, 1116–1122 (2019)], the influence of in-plane magnetic field rotations on the tunnelling behaviour of thin-layer CCPS is negligible.

In response to the Reviewer's inquiry, we have performed in-plane angle-dependent TMR measurements for a sextuple-layer MTJ (6L-S37). As shown in the figure below, we have rotated the magnetic field within the basal plane of 6L-S37 for a full 360-degree cycle in a step of 15 degrees. From the data we can find that H_{sat} remains unchanged during the magnetic field rotations, confirming a negligible influence of the in-plane magnetic field direction. **Note that the in-plane magnetic field rotation results also confirm the conclusion that thin-layer CCPS retain the bulk magnetic properties.**

Related changes to the revised manuscript are the following:

- I. We have added new Figure S17 into the Supplementary Information to show nearly identical TMR results of CCPS 6L-S37 under in-plane H rotations.

Figure S17. TMR measurements of CCPS 6L-S37 under in-plane H rotations. a, Measured TMR versus H with different in-plane angle setpoints. b, dI/dH analyses of the TMR spectra, revealing identical behaviour for different angle setpoints. c, d^2I/dH^2 analyses of the TMR spectra, confirming the dI/dH analyses. d. Polar plot of H_{sat} versus in-plane H angle, proving the easy-plane magnetocrystalline anisotropy of thin-layer CCPS.

Comment 2-13. Why are the work function of the top and bottom graphene electrodes different?

Reply 2-13: Two factors contribute to the work function differences between the top and bottom graphene electrodes. First, as a Dirac system, graphene is well known to be extremely sensitive to environmental doping introduced by sample preparation and device fabrications procedures, as reported by numerous literatures such as [Nat. Mater. 6, 652–655 (2007)], [Phys. Rev. Lett. 98, 076602]. Equally important, the graphene/antiferroelectric CCPS interface will also introduce strong doping to the graphene electrodes, similar graphene/ferroelectric heterostructures have been explored systematically by Yi Zheng et al, see [Appl. Phys. Lett. 94, 163505 (2009)] and [Phys. Rev. Lett. 105, 166602 (2010)] etc.

Strong interfacial doping to graphene by thin-layer CCPS is evident by scanning Kelvin probe microscope, with representative results attached below. As shown in the Figure below, for the tunneling device with an octuple-layer CCPS, the bottom (top-left corner) and top (central area) FLG electrodes exhibit distinctive work functions of about 4.6 eV and 4.3 eV, respectively. The work function differences between top and bottom FLG electrodes are also confirmed by the FN tunnelling method. In Supplementary Note 4, we

show the FN tunnelling fitting of two MTJs, yielding 0.3 eV and 0.55 eV work function differences respectively.

Related changes to the revised manuscript are the following:

- I. Main Text, Page 8, Paragraph 2, Line 12, We have added the sentences: “Using Fowler-Nordheim (FN) tunnelling in complementary with scanning Kelvin probe microscope (SKPM), we determined that the asymmetric work functions can readily reach 0.5 eV due to the Dirac electronic structure of graphene and the interfacial doping by AFE CCPS flakes (see SI Note 4 and Note 8 for detailed analysis using FN tunnelling and SKPM). As shown in SI Fig. 7a, the existence of a build-in potential makes the two otherwise V_b -polarity equivalent FiE states asymmetric in energy, in analogy to GFeFETs under a constant electrostatic doping.”
- II. We have added the Supplementary Note 8 and Figure S12 to show the strong interfacial doping to graphene by thin-layer CCPS.

Figure S12. **Work function measurement of a CCPS device. Left.** Optic image of the device. **Right.** SKPM result of the tunneling device.

Comment 2-14. How many layers of the graphene electrodes was used? Is the work function dependent on the layer number?

Reply 2-14: In our MTJ devices, we mainly use **graphene electrodes of 4-6 monolayers, which readily form narrow ribbon structures ideal for minimizing the effective tunneling junction areas.**

As we have elaborated in Reply 2-13, **the work functions of graphene electrodes are extrinsic and controlled by environmental doping and interfacial doping** from the graphene/antiferroelectric CCPS heterostructure. Both mechanisms contribute strong charge doping in graphene electrodes, making the work functions nearly independent on the layer numbers. Note that the **intrinsic (undoped) work function of graphene, as defined by the energy position of the Dirac point**, is dependent on layer number, see [Diamond and Related Materials, 101, 107576] and [Phys. Rev. B 79, 125437].

Comment 2-15. Dose the interface such as trapped impurities lead to the asymmetric tunneling current?

Reply 2-15: We thank the Reviewer for raising the question. **We fabricate our thin-layer MTJ devices utilizing the well-established methodology** for studying 2D materials, which has proven its validity and efficiency in producing interfacial contamination-free device structures, see [Appl. Phys. Lett. 105, 013101 (2014)] and [Science 360, 1214 (2018)] and numerous literatures on this topic. For the reference of the Reviewer, the step-by-step device fabrication procedures are illustrated in the schematic attached below.

In brief, based on the Polycarbonates (PC) film based dry-transfer technique, the BN/graphene/CCPS/graphene/BN structure are sequentially picked up and assembled into multiple heterostructure devices, during which the interfaces between graphene and thin-layer CCPS are made by stacking freshly exfoliated surfaces, and thus, avoid trapping external contaminations or organic solvent. After the dry-transfer process, the finished device is annealed at 170 °C in the nitrogen filled glove box, which helps to remove any contamination residuals by thermally driving impurities out of the interfacial area.

Comment 2-16. The manuscript should be improved. In Figure 2b, what do z , E , φ_P , P and φ_{AP} stand for? Which parts denote electrodes and CuCrP2S6? Does the spin configuration in the high R and low R state show the same direction?

Reply 2-16: In Figure 2b, label “ z ” stands for the out-of-plane direction perpendicular to the vdW plane, and the tunneling current I_b flows along this direction; “ E ” stands for the electron tunneling energy; “ φ_P ” and “ φ_{AP} ” represent tunneling barrier for spin-up and spin-down electrons, respectively. Using bilayer CCPS as an example, the spin configuration in the high R state is antiparallel, i.e. the stripe-AFE state, and the spin configuration in the low R state is parallel, i.e. the field-aligned FM’ state.

In response to the Reviewer's enquiry, we have made the following changes:

- I. We have updated the labels and added description words for the revised Figure 2b.
- II. We have added new Supplementary Note 2 into the Supplementary Information, explaining in details the spin-filter model of vdW magnets. See: Main Text, Page 4, Paragraph 2, Line 14, “..Figure 2b (see SI Note 2 for detailed information on the spin-filter mechanism).”; and Supplementary Information, Page 1-2, Supplementary Note 2, “Because of the intralayer ferromagnetism and interlayer antiferromagnetic coupling, the energy barrier for opposite-spin tunnelling electrons in each CCPS ML can be treated independently. In each ML of CCPS the transmission coefficients are T_p and T_{ap} for tunnelling electrons with spin momentums parallel or antiparallel to the majority-spin energy barrier. For bilayer CCPS in the AFM state, the final tunnelling transmission probability is $T_{AFM}=2T_pT_{ap}$, while for the FM' state is $T_{FM'}=2(T_pT_p + T_{ap}T_{ap})$. So, the maximum magnetoresistance is $T_{FM'}/T_{AFM} = (T_pT_p + T_{ap}T_{ap})/T_pT_{ap}$. For $T_p > T_{ap}$, $T_{FM'}/T_{AFM} > 1$, which mean that the tunneling current is larger in the FM' state than that in the AFM state.”

Comment 2-17. What does the inset in Figure 4b represent?

Reply 2-17: The inset in the original Figure 4b represents the energy potential diagram of bilayer CCPS under zero and 1 V/nm electric field, respectively. It is to show that without a build-in potential introduced by the asymmetric work functions of top and bottom graphene electrodes, the response of the equilibrium stripe-AFE state is symmetric to an external \mathbf{E} . The dark blue shaded areas highlight the fact that with the presence of a build-in potential, different energy is required for anti-parallel Cu^+ ions to move the same distance (Δd) of 0.5 Å within the FM cages with opposite V_b -setpoints, which explains the experimental observation of V_b -dependent H_{sat} behavior.

In response to the Reviewer's question, we have combined the energy potential diagram of bilayer CCPS in Figure 4b and the ΔE_{MEC} vs Δd curve in Figure 4c into a new

Figure 4b. The new Figure 4c now better explains the V_b -dependent MEC mechanism by correlating the energy potential curve directly to ΔE_{MEC} . See Page 22, Figure 4 and figure captions for subfigure 4b.

Comment 2-18. The calculation detail of the depolarization field should be provided. How does the FiE transition minimize the depolarization effect (page 2)?

Reply 2-18: We agree with the Reviewer that the vdW screening effect is unique and the related calculation details should be presented in the manuscript. Due to vdW layered structure, in which each monolayer is charge neutral, the flipping of an AFE stripe domain within different monolayers require disparate energy, because the neighboring AFE monolayers provides an extra screening mechanism for the formation of a local dipole.

Using bilayer CCPS as an illuminating example, we demonstrate the vdW depolarization effect by DFT calculations of layer-dependent charge density distribution in

response to AFE stripe domain displacements. **Related changes to the manuscript are the following:**

- I. Main Text, Page 7, Paragraph 1, Line 4-7, we have added the sentences on DFT calculations of the vdW depolarization effect: “Such a unique dipole screening mechanism for vdW ferroelectrics, designated as the vdW depolarization effect hereafter, can be well reproduced by DFT calculations of layer-dependent charge density and local potential distribution in response to AFE stripe domain displacements (see SI Note 5 for details).”
- II. The detailed calculation results for explaining the vdW depolarization effect in bilayer CCPS are added into the Supplementary Information as new Figure S9.

Comment 2-19. “2D limit in vdW monolayers of CuCrP2S6 with inherent ferromagnetism and antiferroelectricity.” (page 2) will confuse the readers, since no monolayer was studied in the manuscript.

Reply 2-19: We thank the Reviewer for this suggestion. In response to the Reviewer’s request, **we have revised the sentence to:** “... at the 2D limit in vdW multiferroic

CuCrP₂S₆ with inherent ferromagnetism and antiferroelectricity". See, Main Text, Page 2, Abstract, Line 8-9.

Comment 2-20. There was no gate electrode used. Therefore, gate-tunable or similar statement would not be used.

Reply 2-20: We thank the Reviewer for this useful advice. We fully agree with the Reviewer that the expression of "gate-tunable" may be inappropriate, since strictly speaking, the V_b -dependent external electric field is directly applied on the CCPS channel, rather than employing an extra dielectric layer in the device structure. **Following the Reviewer's advice, we have replaced the phrase "gate tunable" by " V_b -tunable" or " V_b -dependent". Related changes are summarized as follows:**

- I. Main Text, Page 3, Paragraph 2, Line 16: "gate-tunable FiE transitions" changed to "electric field tunable FiE transitions".
- II. Main Text, Page 8, Paragraph 2, Line 7: "ultra gate tunable" changed to "ultra V_b -tunable".
- III. Main Text, Page 8, Paragraph 2, Line 9: "By elucidating the gate-tunable MEC mechanism" to "After elucidating the V_b -tunable MEC mechanism".
- IV. Main Text, Figure 3, Caption, "Ultra-gate tunability" to "Ultra-wide V_b -tunability".

Comment 2-21. Please carefully check the manuscript. It seems that the blue circle in the top middle panel of Figure 4a was missing. 337 meV was missing too. There is no meta-AFE2 states (page 8) in BL CuCrP₂S₆, and there are only meta-AFE1 and meta-AFE3 in Figure 4a. "along the H direction to from the canted AFM (CAFM) state" (page 4) should be "along the H direction from the canted AFM (CAFM) state".

Reply 2-21: We appreciate the detailed proofreading and useful suggestions from the Reviewer. **For clarity and consistency**, we have now identified and differentiated the different metastable AFE states by the format of "layer number-mAFE-Roman numeral", e.g. 2L-mAFE-I represent the lowest energy metastable AFE state of bilayer CCPS. We have also defined the positive polarity of electric field by adding a black solid arrow representing external E.

In response to the Reviewer's suggestions, we have revised the paper accordingly as summarized below:

- I. Main Text, Page 6, Paragraph 2, Line 15; Page 6, Paragraph 3, Line 9; Page 7, Paragraph 1, Line 1-2; Page 9, Paragraph 1, Line 1; and Page 22, Figure 4 Labels and Captions, different mAFE states are properly named by layer number-prefix and Roman numeral-suffix, such as "2L-mAFE-I, 2L-mAFE-II" and "3L-mAFE-I, 3L-mAFE-II, 3L-mAFE-III".
- II. The lattice model representation of quadruple-layer CCPS is moved into Supplementary Information as Figure S8, in which different mAFE states are named by "4L-mAFE-I, 4L-mAFE-II, 4L-mAFE-III".
- III. Figure 4a Captions, sentences explain the naming rule of the different metastable AFE states, "Here, 2L-mAFE-I means the lowest energy metastable AFE state, created by flipping the anti-parallel Cu⁺ ions of the bottom ML, while 2L-mAFE-II corresponds to the flipping the anti-parallel Cu⁺ of the top ML."

Comment 2-22. Some related works should be cited, such as Nanoscale 2019, 11, 5163, Phys. Rev. Mater. 2023, 7, 033402.

Reply 2-22: We thank the Reviewer for reminding us of the related works. **In response, we have cited these works as new reference [44] and [45].**

Reviewer #3 (Remarks to the Author):

Comment 3-1

The authors have reported on the tunneling magnetoresistance in the MTJs with graphene electrodes and a few-layer multiferroic CuCrP₂S₆ insulating barrier. The observed magnetic and electric tunability is both interesting and potentially important for understanding spin-dependent tunneling behaviors, as well as possible magnetoelectric coupling in 2D systems. However, there are several issues that need clarification before considering its publication.

Reply 3-1: We gratefully thank the Reviewer for the professional reviewing and praise of our work. We have addressed the Reviewer's concerns by revising the paper accordingly, fully taking his/her comments into considerations. The changes are summarized below in a point-to-point fashion.

Comment 3-2. The authors have determined the magnetic phase transition of bilayer MTJs solely from the temperature-dependent tunneling conductance data. Low-dimensional systems typically exhibit lower magnetic ordering temperatures than their bulk counterparts. Authors are encouraged to provide additional (and possibly more direct) experimental evidence to support the magnetic phase transitions and their correlation with the tunneling conductance.

Reply 3-2: We thank the Reviewer for the useful suggestion. **Magnetic tunneling junction-based spectroscopy is now a well-accepted methodology for determining the magnetic ordering and the related external force induced phase transitions in 2D vdW magnets**, e.g. seminal works for studying uniaxial c-axis AFM in CrI₃ [Nature 546, 270 (2017)] and vdW easy-plane AFM in CrCl₃ [Nat. Nanotechnol. 14, 1116 (2019)] from multilayer thicknesses down to the monolayer limit. **Our group also demonstrate that tunneling spectroscopy is a superior probe for resolving complex triaxial anisotropy AFM and revealing unconventional AFM-to-ferrimagnetic metamagnetic phase transitions in few-layer CrOCl** [Small 19, 2300964 (2023)].

However, we fully agree with the Reviewer that the tunneling spectroscopy technique should be used with rigorous experimental verifications. **Regarding the magnetic phase diagram of bilayer CCPS, the conclusion is made systematically by temperature-, layer- as well as angle-dependent tunneling spectroscopy, all consistently proving that thin-layer CCPS down to the bilayer thickness retain the bulk magnetic properties.** First, the transition from the canted AFM state to the magnetic moment aligned

FM' state, manifested as a saturation field of H_{sat} in tunneling spectroscopy, directly reflects the interlayer AFM coupling energy of few-layer CCPS. Thus, by measuring temperature-dependent tunneling spectra and determining the vanishing temperature of H_{sat} , the Néel temperature of thin-layer CCPS can be extracted. To verify the results, we have also carried out layer-dependent tunneling spectroscopy for CCPS MTJs with consecutive thicknesses ranging from 8L to bilayer, all consistently showing a Néel temperature of ~ 31 K. In further, we validate the phase diagram by performing angle-dependent tunneling magnetoresistance measurements. As shown in the figure below, by rotating the magnetic field from the in-plane to the out-of-plane directions, the saturation field H_{sat} changes from 3.3 T to 4.8 T, which is consistent with the easy-plane magnetocrystalline anisotropy of CuCrP_2S_6 (see also Reply 2-12 and Reply 3-5 for angle-dependent tunneling spectroscopy with a full 360-degree in-plane rotation)

In response, we have included the angle dependent tunneling data of CCPS MTJs.

Related changes are summarized below:

- I. TMR measurements of CCPS 6L-S37 under an in-plane H rotation of 360 degrees added into the Supplementary Information as Figure S17.
- II. TMR measurements of CCPS BL-S40 with in-plane to out-of-plane H rotations added into the Supplementary Information as Figure S20.

Figure S17. TMR measurements of CCPS 6L-S37 under in-plane H rotation. a. Measured TMR versus H with different in-plane angle setpoints. **b.** dI/dH analyses of the TMR spectra, revealing identical behaviour for different angle setpoints. **c.** d^2I/dH^2 analyses of the TMR spectra, confirming the dI/dH analyses. **d.** Polar plot of H_{sat} versus in-plane H angle, proving the easy-plane magnetocrystalline anisotropy of thin-layer CCPS.

Comment 3-3. It is recommended to include data depicting the thickness-dependent tunneling conductance vs. temperature.

Reply 3-3: We thank the Reviewer for this useful advice. We fully agree with the Reviewer that more data should be included to make the conclusion more convincing. As we have discussed in Reply 3-2, for 2D CCPS, the temperature-dependent tunneling conductance of different layer numbers reveals essentially same Néel temperature of ~ 31 K.

In response to the advice, we have added the temperature-dependent tunneling data of a 7L device. Related changes are listed as follows:

- I. Main Text, Figure 1f, T-dependent tunneling conductance of a 7L sample 7L-S21 added. See also, Main Text, Page 4, Paragraph 2, Line 3-5, “using a small excitation current of 10 nA and zero magnetic field (H), the tunnelling conductance of CCPS MTJs with different thicknesses consistently show an anomaly at ~ 30 K, in excellent agreement with the bulk T_N .”
- II. Supplementary Information, Figure S18, the 2D magnetic phase diagram of 7L-S21 added.

Comment 3-4. The authors have also claimed the existence of a metamagnetic transition based solely on the TMR experiment, which might not be entirely convincing. To enhance the credibility of this claim, it is necessary to include magnetic data from either bulk or bilayer systems.

Reply 3-4: We thank the Reviewer for the suggestion. In response, we include the magnetic data of bulk CCPS single crystals. As shown in the figure below, bulk CCPS is an antiferromagnetic material with a Néel temperature around 31.5 K. Angle-dependent magnetization curves show that the magnetic moments of CCPS lie within the van der Waals plane. The magnetization saturates at about 6.5 T for $H \perp c$, with a net magnetic moment of $\sim 3.15 \mu_B/\text{Cr}$.

In response to this question, these bulk magnetic data have been added into the Supplementary Information as Figure S19.

Comment 3-5. Additionally, it would be beneficial to address questions about in-plane anisotropy.

How does the magnetic transition differ when an in-plane H-field is applied perpendicular to or parallel with the spin direction?

Reply 3-5:

We thank the Reviewer for this advice. We strongly agree with the Reviewer that the in-plane anisotropy is a useful information for 2D CCPS magnetism study. The intralayer ferromagnetic ordering within a CCPS monolayer has an easy-plane magnetocrystalline anisotropy which lies within the van der Waals basal plane. As we have elaborated in Reply 3-2, **thin-layer CCPS retain the bulk magnetic properties**, which means a rather weak in-plane magnetic anisotropy.

In response to the Reviewer's advice, we have performed in-plane angle-dependent TMR measurements for a 6L-S37 MTJ. As shown in the figure below, we have rotated the magnetic field within the basal plane of 6L-S37 for a full 360-degree cycle in a step of 15 degrees. From the data we can find that H_{sat} remains unchanged during the magnetic field rotations, confirming a negligible influence of the in-plane magnetic field direction.

The related changes are the following:

- I. TMR measurements of CCPS 6L-S37 under an in-plane H rotation of 360 degrees added into the Supplementary Information as Figure S17.

Figure S17. TMR measurements of CCPS 6L-S37 under in-plane H rotation. a. Measured TMR versus H with different in-plane angle setpoints. **b.** dI/dH analyses of the TMR spectra, revealing identical behaviour for different angle setpoints. **c.** d^2I/dH^2 analyses of the TMR spectra, confirming the dI/dH analyses. **d.** Polar plot of H_{sat} versus in-plane H angle, proving the easy-plane magnetocrystalline anisotropy of thin-layer CCPS.

Comment 3-6. The manuscript should provide a clear definition of TMR. In Figure 2(c), the TMR in the paramagnetic state is shown to be positive and increases with increasing H-field. Does this imply that the resistance is higher under high H-field conditions in the PM state?

Is this behavior in line with the suggested model? In addition, TMR in the ordered state, as shown in Figure 3(a), increases as the H-field is raised. This seems to contradict the data in Figure 2(c), where TMR decreases until the field reaches 3.5 T in the ordered state. These discrepancies necessitate a thorough discussion.

Reply 3-6: We thank the Reviewer for reminding us of these issues. The discrepancy between Figure 2c and 2e comes from two different definitions of TMR:

$$\text{TMR} = \frac{R(H) - R(H=0)}{R(H=0)} \times 100\%, \text{ and } \text{TMR} = \frac{I_t(H) - I_t(H=0)}{I_t(H=0)} \times 100\%,$$

which differ with each other by an opposite sign. We totally agree with the Reviewer that the definition of TMR should be clear and self-consistent. Following the convention of tunnelling spectroscopy research in the field of 2D magnetic insulators [Nature 546, 270 (2017); Science 360,1218 (2018) and Nat. Nanotechnol. 14, 1116 (2019)], and we unify the TMR definition with the second formula.

In response, we have revised the paper accordingly, and the changes are summarized below:

- I. Main text, Page 4, paragraph 3, line 2: "...tunnelling magnetoresistance (TMR), defined as: $TMR = [I_t(H) - I_t(0)]/I_t(0) \times 100\%$ "
- II. We have updated Figure 2c and 2e so that they are in consistency with the same TMR definition.

Figure 2. **c**, T -dependent TMR vs H_{\parallel} of BL-S44, showing the dwindling of the CAFM saturation field when approaching T_N . **e**, T -dependent TMR vs H_{\parallel} of quintuple-layer MTJ (5L-S12), showing the same dwindling of H_{sat} when approaching T_N .

Comment 3-7. It would be beneficial if the authors could include the number of layers for each figure, such as "N=2, N=5, N=10".

Reply 3-7: We thank the Reviewer for the useful advice. In response we have added the corresponding layer numbers for each device in Figure 2 and 3. Related changes are the following:

- I. We have added "N=2" in Figure 2a and 2c.
- II. We have added "N=5" in Figure 2e.
- III. We have added "N=8" in Figure 3a.

Comment 3-8. Additionally, a comprehensive description of Figure 2(b) is necessary. The spin filter mechanism is difficult to comprehend from the figure alone, and a more detailed explanation of this mechanism is required in the manuscript.

Reply 3-8: We thank the Reviewer for raising this question. We have improved the schematic diagram of the spin-filter tunneling in Figure 2b with extra labels and description text for clarity. In further, detailed explanations on the spin-filter mechanism are now included in the Supplementary Information as SI Note 2. Related changes are summarized as follows:

- I. **Supplementary Information, new Supplementary Note 2:** "Because of the intralayer ferromagnetism and interlayer antiferromagnetism, the barrier for the tunneling electrons in each CCPS ML can be treated independently. In MLs of CCPS the transmission coefficients are T_p and T_{ap} for tunneling electrons parallel or antiparallel to the barrier. For the bilayer CCPS in the AFM state, the final

tunneling transmission probability is $T_{AFM}=2T_pT_{ap}$, while for the FM state is $T_{FM}=2(T_pT_p + T_{ap}T_{ap})$. So, the magnetoresistance is $T_{FM}/T_{AFM} = (T_pT_p + T_{ap}T_{ap})/T_pT_{ap}$. For $T_p>T_{ap}$, $T_{FM}/T_{AFM}>1$, which mean that the tunneling current is larger in the FM' state than that in the AFM state.”

II. **We have revised the illustration of the spin-filter model in Figure 2b with extra labels and description text.**

Figure 2b. **Revised illustration of the spin-filter model.**

Comment 3-9. On page 5, the term " V_b " is utilized without adequate explanation. Is it referring to the gate voltage? If so, where is this gate voltage applied?

Reply 3-9:

We thank the Reviewer for raising the question. Here the V_b refer to the bias voltage applied to the CCPS MTJ devices. The description of gate-tunable means that “electric field tunable”. Strictly speaking, it is inappropriate to use the phrase of “gate-tunable” since no additional dielectric layer is employed in the device structure. **Following the Reviewer’s advice, we have revised the related phrases:**

- I. Main Text, Page 3, Paragraph 2, Line 16: “gate-tunable FiE transitions” changed to “electric field tunable FiE transitions”.
- II. Main text, Page 5, Paragraph 2, Line 1: “Remarkably, CCPS MTJs show bias voltage (V_b) polarity-dependent H_{sat} ”
- III. Main Text, Page 8, Paragraph 2, Line 7: “ultra gate tunable” to “ultra V_b -tunable”.
- IV. Main Text, Page 8, Paragraph 2, Line 9: “By elucidating the gate-tunable MEC mechanism” to “After elucidating the V_b -tunable MEC mechanism”.
- V. Main Text, Page 3, Figure 3 Captions, “Ultra-gate tunability” to “Ultra-wide V_b -tunability”

Comment 3-10. In Figure 3(c), the tunneling current in the CAFM phase (at $H \approx 7$ T, $V_b = 1.6$ V) appears to have a higher magnitude (~ -1.2 uA) than that ($\sim +0.9$ uA) in the FM phase (at $H \approx 7$ T, $V_b = -2$ V). In other words, the CAFM phase under the strong H-field exhibits higher conductance than the FM phase under the same H-field. This behavior appears inconsistent with the proposed model, which would suggest that the CAFM

phase should exhibit lower conductance compared to the FM phase. This discrepancy requires in-depth discussion.

Reply 3-10: We appreciate the Reviewer for the detailed questions. As we have elaborated in Reply 3-2, the tunneling spectroscopy technique should be used with cautions and verifications. **For example, the absolute values of tunneling current should be not used a criterion to differentiate different magnetic phases.** This is because the overall tunneling probability is dependent on both the **barrier height, in which the multiple spin-filter effect plugs in**, and the **work functions of top and bottom FLG electrodes**. For symmetric work functions, different interlayer magnetic moment configuration (CAFM state) will result in a high resistance state, which will make a transition into a low resistance state for the FM' state.

As we have elaborated in Reply 2-13 and Reply 2-14, **the work functions of top and bottom graphene electrodes are usually asymmetric due to extrinsic environmental doping and interfacial doping from the graphene/antiferroelectric CCPS heterostructure**, both lead to the existence of a build-in potential without applying V_b . When such a build-in potential dominates over the spin-filtering effect, tunneling current can be smaller under a positive V_b than that of a negative V_b , which is schematic demonstrated below.

The related changes are the following:

- I. Main text, Page 8, Paragraph 2, Line 12, We have added the sentences: "Using Fowler-Nordheim (FN) tunnelling in complementary with scanning Kelvin probe microscope (SKPM), we determined that the asymmetric work functions can readily reach 0.5 eV due to the Dirac electronic structure of graphene and the interfacial doping by AFE CCPS flakes (see SI Note 4 and Note 8 for detailed analyses using FN tunnelling and SKPM). As shown in SI Fig. 7a, the existence of a build-in potential makes the two otherwise V_b -polarity equivalent FiE states asymmetric in energy, in analogy to GFETs under a constant electrostatic doping."
- II. We have added the Supplementary Note 8 and Figure S12 to show the strong interfacial doping to graphene electrodes by thin-layer CCPS.
- III. We have added the Supplementary Information with new Figure S13 to show the schematic illustration of build-in potential effects on the tunneling current.

Comment 3-11. In Figure 3(d), a negative V_b yields a negative I_t and vice versa. However, in Figure 3(c) at $H = 0$ T, a negative V_b (-2 V) yields a positive I_t and vice versa. Once again, this inconsistency necessitates proper discussion.

Reply 3-11: We thank the Reviewer for carefully proofreading our manuscript. This is a typo during preparing the schematic illustration. **We have corrected the typo by changing the positive V_b direction in Figure 3c to make it consistent with other figures and the illustration in Figure 4a.**

Comment 3-12. On page 7, the authors assert, “As a direct consequence of ultra-gate tunable ΔE_{MEC} , the E-enforced heterogenous FiE state has drastically reduced J_{\perp} , which changes by nearly 50 % with a moderate E of 0.3 V/nm as shown in Fig. 3.” However, this description does not align with Figure 3, as no “ ΔE_{MEC} ” is provided. Moreover, the data exhibiting “the E-field dependent change of J_{\perp} value” is not apparent in any of the figures. These data are crucial for drawing conclusions and should be addressed.

Reply 3-12: We thank the Reviewer for raising the questions. To quantitatively modeling the magnetoelectric coupling, we define a parameter: $\Delta E_{MEC} = (E_{FM}(\mathbf{E}) - E_{AFM}(\mathbf{E})) - (E_{FM}(0) - E_{AFM}(0))$ as a function of external electric field \mathbf{E} . This parameter means the energy difference of thin-layer CCPS between the stripe-AFM ground state to the FM' state, under different \mathbf{E} setpoints. In the 1D spin-chain model [see Rev. Mod. Phys. **90**, 015005 (2018) and Nat. Nanotechnol. **14**, 1116–1122 (2019)], the free energy of an AFM system with an anisotropy constant (K) and subjected to a magnetic field (H) is given by,

$$E = J_{AF} M_s^2 \cos 2\phi + K \cos^2 \phi - \mu_0 H M_s \cos \phi ,$$

in which ϕ represents the CAFM canting angle, J_{AF} (J_{\perp}) is the interlayer AFM coupling energy, and M_s is the net magnetic moment of an individual monolayer. Without considering the Zeeman energy, the free energy of the AFM ground state and the momentum aligned FM' state are,

$$E_{AFM} = J_{AF}M_s^2 \cos(2\pi) + K \cos^2(\pi) = -J_{AF}M_s^2 + K, \text{ and}$$

$$E_{FM'} = J_{AF}M_s^2 \cos(0) + K \cos^2(0) = J_{AF}M_s^2 + K.$$

So, the total energy difference compensated by the Zeeman coupling for the AFM-FM' transition is,

$$E_{FM'} - E_{AFM} = 2J_{AF}M_s^2,$$

which corresponds to a saturation field $H_{\text{sat}} = \frac{2J_{AF}}{\mu_0 M_s}$. Note that the above relation applies

only for the bulk case. For few-layer CCPS, H_{sat} are derived by solving the $N \times N$ matrix of the 1D spin-chain model [*Nat. Nanotechnol.* **14**, 1116–1122 (2019)], and the resulting Layer-dependent CAFM saturation field is:

$$H_{\text{sat}} = \frac{4J_{\perp}}{\mu_0 M_s} \cos^2 \frac{\pi}{2N}.$$

Finally, we get the relation of ΔE_{MEC} ,

$$\Delta E_{\text{MEC}} = (E_{FM'}(\mathbf{E}) - E_{AFM}(\mathbf{E})) - (E_{FM'}(0) - E_{AFM}(0)) = \frac{1}{2 \cos^2 \frac{\pi}{2N}} \mu_0 M_s^3 (H_{\text{sat}}(\mathbf{E}) - H_{\text{sat}}(0)).$$

In conclusion, **both ΔE_{MEC} and J_{\perp} are proportional to V_b -dependent H_{sat} , which is an experimental measurable quantity. So, by determining the H_{sat} values for opposite V_b -polarities, we deduce the change ratio of ΔE_{MEC} as well as J_{\perp} .**

In response to the Reviewer's question, we have updated the manuscript with the following changes:

- I. Main Text, new Figure 3c, showing representative $\Delta E_{\text{MEC}}(\mathbf{E})$ vs V_b data extracted from the TMR data of 10L-S39.
- II. The aforementioned discussions on $\Delta E_{\text{MEC}}(\mathbf{E})$ are added into the Supplementary Information as the Supplementary Note 7.

Comment 3-13. On page 8, the authors mention, “The striking energy... the meta-AFE1 and meta-AFE2 states of BL CCPS...”. However, as demonstrated in Figure 4 (the first

row), there is no meta-AFE2 state in BL. Shouldn't it be meta-AFE3 state instead? The terms "meta-AFE2,3" need to be precisely defined in the manuscript.

Reply 3-13: We appreciate the detailed proofreading from the Reviewer. **For clarity and consistency**, we have now identified and differentiated the different metastable AFE states by the format of "layer number-mAFE-Roman numeral", e.g. 2L-mAFE-I represent the lowest energy metastable AFE state of bilayer CCPS. We have also defined the positive polarity of electric field by adding a black solid arrow representing external E.

In response to the Reviewer's question, we have revised the paper which is summarized as follows:

- I. Main Text, Page 6, Paragraph 2, Line 15; Page 6, Paragraph 3, Line 9; Page 7, Paragraph 1, Line 1-2; Page 9, Paragraph 1, Line 1; and Page 22, Figure 4 Labels and Captions, different mAFE states are properly named by layer number-prefix and Roman numeral-surfix, such as "2L-mAFE-I, 2L-mAFE-II" and "3L-mAFE-I, 3L-mAFE-II, 3L-mAFE-III".
- II. The lattice model representation of quadruple-layer CCPS is moved into Supplementary Information as Figure S8, in which different mAFE states are named by "4L-mAFE-I, 4L-mAFE-II, 4L-mAFE-III".
- III. Figure 4a Captions, sentences explain the naming rule of the different metastable AFE states, "Here, 2L-mAFE-I means the lowest energy metastable AFE state, created by flipping the anti-parallel Cu⁺ ions of the bottom ML, while 2L-mAFE-II corresponds to the flipping the anti-parallel Cu⁺ of the top ML."

Comment 3-14. On page 3, the authors declare, "Here, we reported ... tunable 2D MEC... fundamentally different from the bulk counterparts." However, recent researches have observed E-field induced local polarization and significant magnetoelectric coupling in bulk samples (Adv. Funct. Mater. 2022, 32, 2204214 & Adv. Electron. Mater. 2022, 2101072). The authors must carefully review recent progress in the bulk system to ensure accurate statements.

Reply 3-14: We thank the Reviewer for this expert suggestion. We have included these two excellent works on bulk CCPS into the revised manuscript. **Related changes are the following:**

- I. Main Text, Page 13, new references [48] and [49].

Again, we would like to thank all the Reviewers for their thoughtful comments and suggestions, which we think have helped to greatly improve the readability and clarity of our manuscript.

REVIEWER COMMENTS

Reviewer #1 (Remarks to the Author):

The authors have addressed all the issues. I suggest to accept.

Reviewer #2 (Remarks to the Author):

The authors have successfully addressed most of my concerns. And the manuscript can be published after solving the following small comment.

For SHG measurements, the authors are encouraged to show the polarization-dependent SHG optical path patterns at different temperature and V_b in the main text, as they can directly reflect the symmetries. Meanwhile, the detailed set-up (not only the set-up for the device) for the SHG measurements need to be provided in the main text.

Reviewer #3 (Remarks to the Author):

The authors have addressed many of the concerns raised by reviewers and responded to their comments. However, new data once again raises further questions and does not resolve the important issues raised by referees. Additionally, there are still issues that require further clarification and corrections. Therefore, I do not agree with the publication of the manuscript at the current stage.

1. Tunnelling magnetoresistance (TMR):

A. In Figure 2, the saturation magnetic field (H_{sat}) is 3 T for $N=2$ and 5.5 T for $N=5$ without bias voltage ($V_b=0$). However, in Figure 3, the $N=8$ sample shows $H_{sat}=8.8$ T with positive V_b and $H_{sat}=5.9$ T with negative V_b . Both bias voltages result in higher H_{sat} than the $H_{sat}=3$ T in $V_b=0$. Are these discrepancies due to the different number of layers in the sample? On the other hand, Figure 1f indicates no thickness dependence for the magnetic transition temperature, T_N . Systematic analyses for thickness dependence are required.

B. In Figure S19, magnetization versus field (MH) data on the bulk sample indicates $H_{sat} = 6.5$ T. However, thin layered CCPS measured by TMR is claimed to show $H_{sat}=3.2$ T in $N=2$ or 5.5 T in $N=5$ samples. Therefore, the bulk and thin layered samples exhibit different H_{sat} values. This point should be discussed to verify if the TMR can determine the intrinsic magnetic phase transition.

2. Second harmonic generation (SHG):

A. The new SHG data in Figure 4 does not serve as experimental proof for either the V_b induced displacement of Cu ions or the existence of FiE phase. How is the change in inversion symmetry breaking related to the intensity change of SHG experiment?

B. Why does the intensity change of SHG show asymmetry upon V_b ? Can the different work function of the electrodes, not the FiE phase transition, generate the intensity change of SHG signal?

3. Other required corrections:

A. Page 2, line 21: "... and type-II multiferroics driven by the Dzyaloshinskii-Moriya interaction". Type-II multiferroics are categorized not only by the DM interaction but also by

other types of magnetic orderings. See Khomskii et al., *Physics* 2, 20 (2009) and Cheong et al., *Nature Materials*, 6, 13 (2007).

B. Page 6, line 19: “the displacement (d) of the anti-parallel stripe domain”. Is it the displacement of domain itself or the vertical displacement of Cu ions within the unit cell?

C. Figure 4e: Two energy diagram cartoons look identical. It is suggested to add words such as “Strong coupling” and “Weak coupling” on the graph.

D. Page 8, line 15: “As summarized in Fig. 4c,” should be “4b”

E. Page 8, line 23: “50% with a moderate E of 0.3 V/nm as shown in Fig. 3” should be “Fig. 3(c)”.

F. Authors are required to carefully proofread their manuscript. If an abbreviation is used, it should be properly defined at the first place. For example, on page 5, line 6, “a quintuple-layer MTJ (5L-S5).” What does the “S5” represent?

REVIEWER COMMENTS

Reviewer #2 (Remarks to the Author):

The authors have successfully addressed most of my concerns. And the manuscript can be published after solving the following small comment.

For SHG measurements, the authors are encouraged to show the polarization-dependent SHG optical path patterns at different temperature and V_b in the main text, as they can directly reflect the symmetries. Meanwhile, the detailed set-up (not only the set-up for the device) for the SHG measurements need to be provided in the main text.

Reply 2-1: We thank Reviewer for these insightful suggestions. The polarization dependent SHG measurement is indeed a direct reflection of the underlying crystal symmetry, and provides an intuitive visualization on how the electric field breaks the symmetry by shifting Cu^+ ion from anti-parallel AFE positions.

Following Reviewer's advice, we have conducted polarization-dependent SHG measurements at 300 K and 77 K under different V_b setpoints with a graphene/CCPS/BN/graphene device structure. As shown in the Figures attached below, the polarization-dependent SHG results unambiguously demonstrate the highly effective gate-tunability of Cu^+ ion displacements by external V_b . Uniquely, by entering the AFE state, each CCPS monolayer forms a distinctive mirror symmetry along the b -axis, in perpendicular to the Cu^+ AFE stripe chains (see Figure 1a; Note that the b -axis also defines the interlayer stacking direction between two neighboring monolayers for the formation of a double-layer unit). By applying V_b , the anti-parallel Cu^+ ions are rearranged within the CuS_6 vdW cages, which effectively break the double-layer inversion symmetry and enhance the SHG signals along the b -axis, while the SHG enhancements perpendicular to the b -axis are significantly weaker due to the mirror symmetry. Consequently, the polarization-dependent SHG evolve from a six-fold-like geometry at 300 K to a two-fold pattern at 77 K. As shown in the Supplementary Fig. 14c for 300 K and the new Figure 4d for 77 K respectively, both are also attached below for the reference of the Reviewer, the SHG patterns are highly repeatable for continuous voltage ramping up and down. These results confirm that the changes of SHG signals, manifesting optical anisotropy continuously tuned by different V_b setpoints, originate from Cu^+ rearrangements associated with the AFE to FiE transitions of CCPS. Note that V_b used in the polarization-dependent experiments is larger than that in the previous SHG measurement, due to the variations of device parameters, especially the thicknesses of the BN layer. Equally important, it is well known that the optical non-linearity is quadratically dependent on the electric field of the laser light. For polarization-dependent SHG measurements, which require a much longer data acquisition time than the standard SHG experiments, we utilized laser power at **5 mW** and centered excitation wavelength at **1560 nm**, which is below the bandgap to minimize direct photon absorption for the protection of the CCPS channels from overheating damage.

We also want to emphasize that these new polarization-dependent results are essentially the same as the previous present SHG data in Figure 4d of the first revised version, which are technically the angle-integration of polarization-dependent SHG intensity under different V_b setpoints.

In response to the Reviewer's suggestions, we have updated the manuscript accordingly. The related changes are listed below:

- I.** We have added polarization-dependent SHG results at 77 K with different V_b setpoints in Figure 4d of the main text.
- II.** We have added polarization-dependent SHG data at 300 K with different V_b in the Supplementary Fig.14c.
- III.** We have added the SHG experimental information such as the optical setup and device fabrications in the method section of the main text and in Supplementary Fig.14d.

REVIEWER COMMENTS

Reviewer #3 (Remarks to the Author):

The authors have addressed many of the concerns raised by reviewers and responded to their comments. However, new data once again raises further questions and does not resolve the important issues raised by referees. Additionally, there are still issues that require further clarification and corrections. Therefore, I do not agree with the publication of the manuscript at the current stage.

1. Tunnelling magnetoresistance (TMR):

A. In Figure 2, the saturation magnetic field (H_{sat}) is 3 T for $N=2$ and 5.5 T for $N=5$ without bias voltage ($V_b=0$). However, in Figure 3, the $N=8$ sample shows $H_{\text{sat}}=8.8$ T with positive V_b and $H_{\text{sat}}=5.9$ T with negative V_b . Both bias voltages result in higher H_{sat} than the $H_{\text{sat}}=3$ T in $V_b=0$. Are these discrepancies due to the different number of layers in the sample? On the other hand, Figure 1f indicates no thickness dependence for the magnetic transition temperature, T_N . Systematic analyses for thickness dependence are required.

B. In Figure S19, magnetization versus field (MH) data on the bulk sample indicates $H_{\text{sat}} = 6.5$ T. However, thin layered CCPS measured by TMR is claimed to show $H_{\text{sat}}=3.2$ T in $N=2$ or 5.5 T in $N=5$ samples. Therefore, the bulk and thin layered samples exhibit different H_{sat} values. This point should be discussed to verify if the TMR can determine the intrinsic magnetic phase transition.

Reply 3-1A: We appreciate the Reviewer for raising up these critical questions. **The monolayer number-dependent metamagnetic saturation fields H_{sat} and thickness-independent Néel temperature T_N are well expected for 2D magnetic materials with intralayer FM and interlayer AFM coupling.** In general, the formation of the AFM ground state, as parameterized by T_N , is predominated by the intralayer FM exchange interactions (J_{\parallel}), while the interlayer AFM coupling (J_{\perp}), determines the external magnetic coupling energy required to spin canting the interlayer AFM phase, as represented by H_{sat} . It should be pointed out that by reaching H_{sat} , each CCPS monolayer remains the intralayer FM ordering as determined by J_{\parallel} , while the magnetization directions of different monolayers become aligned since J_{\perp} is compensated by the external magnetic coupling energy.

The monolayer number-dependent H_{sat} has been systematically studied in the literatures such as the archetypical CrCl_3 system [*Nat. Nanotechnol.* 14, 1116–1122 (2019)]. In brief, the layer-dependent H_{sat} are derived by solving the $N \times N$ matrix of the 1D spin-chain model, in which N is the layer number. Quantitatively, the calculations for H_{sat} , which is already discussed in details in the method section of the main text, are based on the following canted AFM formula:

$$H_{\text{sat}} = \frac{4J_{\perp}}{\mu_0 M_S} \cos^2 \frac{\pi}{2N} \quad (1).$$

Apparently, the H_{sat} value is determined by both interlayer exchange strength J_{\perp} and the layer number N . As we have elaborated in the previous paragraph, once below the AFM transition T_N , CCPS develops a fixed interlayer exchange strength J_{\perp} , while H_{sat} varies in different-layer samples. So, the layer-dependence of H_{sat} is the intrinsic property of a vdW AFM magnet with intralayer FM and interlayer AFM ordering, and it should not be interpreted as the equivalence of the AFM Néel temperature T_N .

On the other hand, due to the unique vdW layered structure of 2D AFM magnets, the interlayer magnetic coupling J_{\perp} is typically much weaker than the intralayer coupling strength J_{\parallel} . Such a magnetic anisotropy plays a decisive role in the formation of 2D magnetism, making T_N predominantly depend on J_{\parallel} , which is nearly constant for different samples from multilayer to few-layer thicknesses. Furthermore, in our magnetoelectric coupling experiments of CCPS, we realize the electric control of J_{\perp} by lattice distortions introduced by Cu^+ ions displacements within the CuS_6 vdW cages, which lower the interlayer superexchange interaction, but hardly affect the in-plane coupling J_{\parallel} and hence T_N . So, there is no discrepancy at all to get the same T_N for different layer-number samples while observing the layer-dependent H_{sat} .

Reply 3-1B: By clarifying the different physical meaning of H_{sat} and T_N in Reply 3-1A, it is straightforward to estimate the H_{sat} values for the bulk and few-layer samples by Eq. (1).

In particular, for bilayer samples, the result is $H_{\text{sat}}(2\text{L}) = \frac{4J_{\perp}}{\mu_0 M_s} \cos^2 \frac{\pi}{4} = \frac{2J_{\perp}}{\mu_0 M_s}$, while in the

bulk limit where $N \rightarrow \infty$, $H_{\text{sat}}(\text{bulk}) = \frac{4J_{\perp}}{\mu_0 M_s} \cos^2 0 = \frac{4J_{\perp}}{\mu_0 M_s}$. So, the canted AFM model gives a precise prediction on $H_{\text{sat}}(\text{bulk}) = 2H_{\text{sat}}(2\text{L})$, which is perfectly supported by our measured data of $H_{\text{sat}}(\text{bulk}) = 6.5$ T and $H_{\text{sat}}(2\text{L}) = 3.2$ T.

Note that for multilayer CCPS devices under V_b , the H_{sat} values are effectively tuned by the external electric field which induces continuous AFE-to-FiE (ferrielectricity) transitions. Nevertheless, we can see that $H_{\text{sat}}(5\text{L}) = H_{\text{sat}}(\text{bulk}) \cos^2 \frac{\pi}{10} = 5.8$ T still gives an excellent prediction on the experimental observation of 5.5 T for the $N=5$ device in Figure S19.

2. Second harmonic generation (SHG):

A. The new SHG data in Figure 4 does not serve as experimental proof for either the V_b induced displacement of Cu ions or the existence of FiE phase. How is the change in inversion symmetry breaking related to the intensity change of SHG experiment?

B. Why does the intensity change of SHG show asymmetry upon V_b ? Can the different work function of the electrodes, not the FiE phase transition, generate the intensity change of SHG signal?

Reply 3-2A: We thank the Reviewer for raising these professional questions. By probing the second order non-linear optical response of a material system, the SHG technique is a well-established tool to identify potential phase changes and resolve the resulting crystal structures with unprecedented sensitivity and versatility to be used with external modulation forces, e.g. temperature, polarization and external fields etc. SHG has also

been widely used for studying 2D ferroelectric materials such as In_2Se_3 and CuInP_2S_6 [Phys. Rev. Lett. 120, 227601 (2018); Nat. Comm. 7, 12357 (2016)].

In principle, as a nonlinear optical method, the SHG signals can only be observed in crystal systems lacking of inversion symmetry, which in CCPS crystals are largely controlled by the Cu^+ ions arrangements within the CuS_6 vdW cages. To make our SHG results on confirming the FiE phases more intuitive to see, **we have conducted polarization-dependent SHG measurements at 300 K and 77 K**, respectively, under different V_b with a graphene/CCPS/BN/graphene device structure. Uniquely, by entering the AFE state below 145 K, each CCPS monolayer forms a distinctive mirror symmetry along the b-axis, in perpendicular to the Cu^+ AFE stripe chains (see Figure 1a; Note that the b-axis also defines the interlayer stacking direction between two neighboring monolayers for the formation of a double-layer unit). By applying V_b , the anti-parallel Cu^+ ions are rearranged within the CuS_6 vdW cages, which effectively break the double-layer inversion symmetry and enhance the SHG signals along the b-axis, while the SHG enhancements perpendicular to the b-axis are significantly weaker due to the mirror symmetry. Consequently, the polarization-dependent SHG evolve from a six-fold-like geometry at 300 K to a two-fold pattern at 77 K. As shown in the Supplementary Fig. 14c for 300 K and the new Figure 4d for 77 K respectively, both are also attached below for the reference of the Reviewer, the SHG patterns are highly repeatable for continuous voltage ramping up and down. These results unambiguously confirm that the changes of SHG signals, manifesting optical anisotropy continuously tuned by different V_b setpoints, originate from Cu^+ rearrangements associated with the AFE to FiE transitions of CCPS.

We want to emphasize that these new polarization-dependent results are essentially the same as the previous present SHG data in Figure 4d of the first revised version, which are technically the angle-integration of polarization-dependent SHG intensity under different V_b setpoints. See also **Reply 2-1** for details on polarization-dependent SHG.

In response to these suggestions, we have updated the manuscript accordingly. The related changes are summarized as follows:

- I. We have added polarization-dependent SHG results at 77 K with different V_b setpoints in Figure 4d of the main text.
- II. We have added polarization-dependent SHG data at 300 K with different V_b in the Supplementary Fig.14c.
- III. We have added the SHG experimental information such as the optical setup and device fabrications in the method section of the main text and in Supplementary Fig.14d.

Reply 3-2B: As we have elaborated in the main text for explaining the bias-asymmetric magnetic tunneling data, the SHG intensity asymmetry upon V_b originates from the same built-in potential introduced by the different work functions of top and bottom FLG electrodes. As described in the Reply 3-2A, the SHG measurement is based on nonlinear optical anisotropy of a crystalline system, which can only be observed in crystal systems lacking of inversion symmetry. In the case of few-layer and multilayer CCPS, the inversion symmetry is controlled by V_b -dependent Cu^+ ions rearrangements within the CuS_6 vdW cages, which is unique for the CCPS lattice. Indeed, **for control experiments on BN, FLG electrodes or SiO_2 areas on the substrates, neither temperature-dependent SHG evolution (Figure 4c of the main text), nor the fingerprinting six-fold to two-fold symmetry transition (Supplementary Fig. 14c and new Figure 4d respectively) have been observed.**

3. Other required corrections:

Comment 3-3A. Page 2, line 21: "... and type-II multiferroics driven by the Dzyaloshinskii-Moriya interaction". Type-II multiferroics are categorized not only by the DM interaction but

also by other types of magnetic orderings. See Khomskii et al., *Physics* 2, 20 (2009) and Cheong et al., *Nature Materials*, 6, 13 (2007).

We sincerely thank the Reviewer for carefully proofreading of our manuscript and for these useful suggestions.

Reply 3-3A: We agree with the Reviewer that a more comprehensive description of type-II multiferroics should be stressed in the article. In the reference [Khomskii et al., *Physics* 2, 20 (2009)], they generalized the conventional type-II multiferroic into two sub-groups: “those in which ferroelectricity is caused by a particular type of magnetic spiral and those in which ferroelectricity appears even for collinear magnetic structures.”

In response to the Reviewer’s suggestion, we have improved the related sentences as: “... and type-II multiferroics originated from Dzyaloshinskii-Moriya interaction and other types of magnetic orderings such as collinear magnetic structures [3-5].” And we have added two new references, [*Physics* 2, 20 (2009)] and [*Nature Materials*, 6, 13 (2007)] respectively, in the main text.

Comment 3-3B. Page 6, line 19: “the displacement (d) of the anti-parallel stripe domain”. Is it the displacement of domain itself or the vertical displacement of Cu ions within the unit cell?

Reply 3-3B: Here the “the displacement (Δd) of the anti-parallel stripe domain” means the displacements of Cu⁺ ions along the vertical direction within the CuS₆ vdW cages.

In response to the Reviewer’s suggestion, we have improved the related sentences to: “...the displacement (Δd) of the anti-parallel Cu⁺ ion stripes...”

Comment 3-3C. Figure 4e: Two energy diagram cartoons look identical. It is suggested to add words such as “Strong coupling” and “Weak coupling” on the graph.

Reply 3-3C: We agree with the Reviewer’s suggestion and we have updated the dashed double-headed arrows in Figure 4e with distinctive line width, with additional captions of “Strong coupling” and “Weak coupling” to highlight the weakening of interlayer superexchange coupling induced by the AFE-FiE transitions. The updated Figure 4e is also attached below for the reference of the Reviewer.

Comment 3-3D. Page 8, line 15: “As summarized in Fig. 4c,” should be “4b”

Reply 3-3D: We thank the Reviewer for carefully proofreading, and we have revised the typo in this revision.

Comment 3-3E. Page 8, line 23: “50% with a moderate E of 0.3 V/nm as shown in Fig. 3” should be “Fig. 3(c)”.

Reply 3-3E: We gratefully thank the Reviewer for carefully proofreading the manuscript, and we have revised the typo in this revision.

Comment 3-3F. Authors are required to carefully proofread their manuscript. If an abbreviation is used, it should be properly defined at the first place. For example, on page 5, line 6, “a quintuple-layer MTJ (5L-S5).” What does the “S5” represent?

Reply 3-3F: Here the notation for our tunnelling devices is “Layer number”-“sample serial number”. So here “5L-S5” means the 5th device which has a five-layer CCPS channel, while “BL-S44” stands for the 44th sample which has a bilayer CCPS channel. Note that for each CCPS thickness, the results have been verified and repeated for at least three devices.

REVIEWERS' COMMENTS

Reviewer #2 (Remarks to the Author):

The manuscript can be published since the authors have successfully addressed my concerns.

Reviewer #3 (Remarks to the Author):

I am satisfied with the authors' response and, therefore, agree with the publication of the manuscript in Nature Communications.